# Weather extremes over Europe under 1.5°C and 2.0°C global warming from HAPPI regional climate ensemble simulations

Kevin Sieck[1], Christine Nam[1], Laurens M. Bouwer[1], Diana Rechid[1], Daniela Jacob[1]

[1]Climate Service Center Germany (GERICS), Helmholtz-Zentrum Geesthacht, Hamburg, 20095, Germany

*Correspondence to*: Kevin Sieck (kevin.sieck@hzg.de)

**Abstract.** This paper presents a novel dataset of regional climate model simulations over Europe that significantly improves our ability to detect changes in weather extremes under low and moderate levels of global warming. This is a unique and physically consistent dataset, as it is derived from a large ensemble of regional climate model simulations. These simulations were driven by two global climate models from the international HAPPI consortium. The set consists of 100 x 10-year simulations and 25 x 10-year simulations, respectively. These large ensembles allow for regional climate change and weather extremes to be investigated with an improved signal-to-noise ratio compared to previous climate simulations. To demonstrate how adaptation-relevant information can be derived from the HAPPI dataset changes in four climate indices for periods with 1.5°C and 2.0°C global warming are quantified. These indices include number of days per year with daily mean near-surface apparent temperature of >28°C (ATG28); the yearly maximum 5-day sum of precipitation (RX5day); the daily precipitation intensity of the 50-yr return period (RI50yr); and the annual Consecutive Dry Days (CDD). This work shows that even for a small signal in projected global mean temperature, changes of extreme temperature and precipitation indices can be robustly estimated. For temperature related indices changes in percentiles can also be estimated with high confidence. Such data can form the basis for tailor-made climate information that can aid adaptive measures at policy-relevant scales, indicating potential impacts at low levels of global warming at steps of 0.5°C.

## 1 Introduction

Identifying regional climate change impacts for different global mean temperature targets is increasingly relevant to both the private and public sector. In the private sector, investors demand financial disclosure associated with climate change risks and opportunities (Goldstein, et al., 2018). In the public sector, policy makers rely on climate information build on internationally agreed limits to develop national climate action policies. This is especially true after the adoption of the Paris Agreement of the United Nations, which aims to keep global climate warming well below 2.0°C compared to pre-industrial times (UNFCCC, 2015). Temperature targets, however, are not directly related to the representative concentration pathways (Van Vuuren et al., 2011) used in the generation of global climate simulations (CMIP5, Taylor et al., 2012) and the Shared Socioeconomic Pathways (Meinshausen et al., 2019) used in CMIP6 (Eyring et al., 2016). Therefore, new techniques are being developed to extract information on the possible implications of further global warming. Recent studies using CMIP5 data have shown that

climate change indices can be extracted for different warming levels, by identifying specific time periods when a certain global mean temperature (GMT) increase is reached in a general circulation model (GCM) (Schleussner et al., 2016; Vautard et al., 2014; Jacob et al., 2018). These studies typically used 5 to 15 ensemble members, which were available in CMIP5 at the time, for their global and regional studies. However, Mitchell et al. (2016) argued that a different experiment design is needed to better address the policy-relevant temperature targets with climate simulations, because the relatively small CMIP5 ensemble does not provide the necessary size to quantify changes in weather extremes at low levels of warming. The high natural variability in models requires the creation of large ensemble datasets (Deser et al., 2013). Following the recommendations of Mitchell et al. (2016), the HAPPI consortium ("Half a degree Additional warming, Prognosis and Projected Impacts") designed targeted experiments created for the purpose of extracting the required information on distinct warming levels using 10 state-of-the-art GCMs (Mitchell et al., 2017). The HAPPI experiments include a large number of ensemble members, typically 50 to 100 members per GCM, using AMIP-style integrations (Gates et al., 1992), which significantly improves the signal-to-noise ratios. A better signal-to-noise ratio is essential for differentiating between impacts from 1.5°C and 2.0°C global warming, especially for changes in weather extremes.

Regional climate impact assessments often require a much higher resolution than GCMs currently have (e.g., Giorgi and Jones, 2009). To bridge this gap, dynamical downscaling with Regional Climate Models (RCMs) is an effective option as they provide physically consistent high-resolution climate information (Jacob et al., 2014; Giorgi and Gutowski, 2015; Gutowski et al., 2016). Here, the RCM REMO (Jacob et al., 2012) is used to dynamically downscale simulations from two GCMs of the HAPPI consortium. Two regional climate datasets of 25 and 100 members are developed to create a large ensemble of RCM simulation, which are particularly suitable to study extremes. Earlier studies such as Leduc et al. (2019) have successfully demonstrated the usefulness of such an approach. To demonstrate the potential of this dataset for regional climate impact studies, under 1.5°C and 2.0°C global warming, changes in four climate indices for weather extremes are quantified.

In Section 2, we present the REMO regional climate model, experiment setup and simulations performed. In Section 3, the changes to four climate indices for extreme weather are derived from the HAPPI dataset. Lastly, the conclusions are presented in Section 4.

## 2 Methods

To create a dataset for regional climate impact studies for Europe under 1.5°C and 2.0°C global warming, the regional climate model REMO has been used to dynamically downscale two GCM ensembles following the HAPPI experiment protocol by Mitchell et al. (2017). The major aspects of the HAPPI experiment protocol are summarized in the following subsection, as there are important differences compared to the typical CMIP protocols. Following which, the RCM experimental set up is introduced. Lastly, several common climate indices are computed to demonstrate the usefulness of this new dataset.

## 2.1 Global HAPPI simulations

The HAPPI protocol by Mitchell et al. (2017) has been set up to inform the IPCC Special Report on 1.5°C Warming (IPCC, 2018). The idea is that large ensembles (>50 members) of GCM simulations will allow extreme events to be studied, even for the small differential warming between a current decade (2006-2015) and two future decades under 1.5°C and 2.0°C global warming.

All simulations were conducted in atmosphere-only mode in order to increase ensemble size. Atmosphere-only mode simulations have lower computational costs (Mitchell et al., 2017) and can provide more accurate regional projections because they do not suffer from systematic biases such as SST drifts (He and Soden, 2016). The simulation period for all members is limited to 10 years, because during the current period from 2006-2015 sea-surface temperatures stayed approximately constant. This period forms the basis of the entire experiment and allows for a better estimate of, e.g., return-values from this period compared to periods with a strong warming trend. The experiment design for the current decade follows the DECK AMIP protocol using observed sea ice and SSTs. For the future periods, SSTs are calculated by taking the 2006-2015 observed conditions and adding a SST increment representing the future periods.

The multi-model averaged CMIP5 global mean temperature response for 2091-2100 compared to 1861-1880 under RCP2.6 is 1.55°C. Mitchell et al. (2017) considered this warming as sufficiently close to inform about impacts under 1.5°C and chose this period under RCP2.6 as basis for a 1.5°C warmer period. The SST anomalies for the 1.5°C period were computed using the modelled decade averaged difference between 2091-2100 from RCP2.6 and 2006-2015 from RCP8.5, as RCP8.5 averaged SSTs over this period are closest to observations. Forcing values for anthropogenic greenhouse gases, aerosols and land-use are taken from the year 2095 of RCP2.6 and kept constant during the simulation. Because of the poor representation of sea ice in the CMIP5 models, Mitchell et al. (2017) used a different approach to construct sea ice concentrations for the 1.5° period. A detailed description is beyond the scope of this paper and can be found in the cited reference.

The SST anomalies for the 2.0°C period cannot be calculated following a similar approach as for the 1.5°C period, because none of the RCPs show a global mean temperature response close to 2.0°C at the end of the century. Therefore, Mitchell et al. (2017) calculated a weighted sum of the RCP2.6 and RCP4.5 multi-model global mean temperature response using the following formula: $w_1 \times T_{RCP2.6} + w_2 \times T_{RCP4.5}$ with $w_1 = 0.41$ and $w_2 = 0.59$. The result adds up to 2.05°C, which is exactly 0.5°C more compared to the modelled warming under RCP2.6 of 1.55°C (see above). The calculation of SST anomalies and sea ice extent follows the same methodology as for the 1.5°C period. Mitchell et al. (2017) decided to apply their weighting method only to the well-mixed greenhouse gases, because the land-use changes and aerosols show very different spatial patterns and are therefore kept at the 1.5°C period values.

## 2.2 Regional HAPPI simulations

In order to create high-resolution climate data for Europe from HAPPI, the RCM REMO has been used for downscaling. REMO is a hydrostatic limited-area model of the atmosphere that has been extensively used and tested in climate change

studies over Europe (Jacob et al., 2012; Teichmann et al., 2013; Kotlarski et al., 2014). The simulation domain follows the CORDEX specification for the standard European domain with 0.44° horizontal resolution. The European CORDEX domain for REMO covers 121x129 grid boxes. To exclude the zone where the REMO simulations are relaxed towards the GCM solutions, a core domain of 106x103 grid boxes, following the CORDEX definition, is used for the analyses. In the vertical, 27 levels are used without nudging except for the boundaries. Boundary conditions are taken from the HAPPI Tier1 experiments (Mitchell et al., 2017), which are carried out with ECHAM6 in T63 (1.875°) horizontal resolution (Stevens et al., 2013; Lierhammer et al., 2017) (100 members per period) and NorESM in 1.25°x0.94° horizontal resolution (Bentsen et al., 2013) (25 members per period). Both models provide 6-hourly 3-dimensional data for downscaling. In REMO the same greenhouse gas forcings as for the GCMs were used and no land-use changes were applied.

SST and sea ice concentrations were taken directly from the GCM output matching the GCM land-sea mask for NorESM. From ECHAM6 only the sea ice concentrations were taken. Due to the interpolation procedure for the sea ice extent, it could happen that sea ice was artificially created where no ice conditions were present in the original dataset, e.g., during summer in the Baltic Sea. ECHAM6 has a mechanism that as soon as there is a fraction of sea ice greater than zero, the SST is limited to a maximum of 272.5K. This leads to artificial temperature jumps in the SST between adjacent grid boxes as soon as erroneous sea ice appeared in one of the grid boxes. In order to avoid inheriting this issue, the originally provided SST fields from the HAPPI project were used for the REMO simulations, using ECHAM6 as forcing GCM. After testing different temperature and/or sea ice fraction thresholds, the authors decided to keep the original sea ice maps, because in cases where artificial sea ice was created the fraction was typically well below 1%, and only in rare cases reaches up to 4% (not shown). All other procedures would have removed too much sea ice in other seasons or led to unrealistic gradients of sea ice fractions. With the tile approach of REMO, the effect of the artificial sea ice on the averaged near-surface variables is hardly detectable.

For each GCM member only one REMO simulation was carried out, as inter-member variability of an RCM ensemble over Europe on a time scale of 10 years is small compared to the internal variability of a GCM (Sieck et al., 2016). Each simulation covers a period of ten years, and as such, initial conditions for the lower boundary need to be in balance with the RCM's internal climate in order to avoid artificial drifts in the modelled results. To achieve this, for each driving GCM, the first year of a random GCM member was simulated five times with REMO using initial conditions from the end of the previous run, creating one initial soil state for every ensemble member in one period. This was performed for each of the three periods. Tests showed that this minimizes drifts in the deep soil climatology compared to initial conditions taken directly from the GCM (not shown).

We performed a qualitative analysis of the results compared to observations. In general, the performance of the HAPPI-ensemble is in-line with typical results from the CMIP-type downscaling activities performed in the CORDEX framework with REMO (see the supplement for figures and discussion). A list of variables and frequencies available from the REMO simulations can be found in the supplement (see Table S-1).

## 2.3 Climate indices

To demonstrate how adaptation-relevant information can be derived from the HAPPI dataset for two different average global temperature targets, four climate indices used in climate impact studies are presented. The extremes are selected based on recommended indices developed by the joint CCI/CLIVAR/JCOMM Expert Team on Climate Change Detection and Indices (ETCCDI) (Karl et al., 1999; Frich et al., 2002) and other indicators. The selected climate indices are: number of days per year with a daily mean near-surface apparent temperature of more than 28°C (ATG28); annual maximum 5-day sum of precipitation (RX5day); change in daily precipitation intensity at the 50-yr return period (RI50yr); consecutive dry days (CDD) as a measure of meteorological drought.

All four climate indices are calculated for each year and ensemble member. With 100 and 25 members, respectively, for 10 years each, the ECHAM6 driven ensembles yield 1000 data points for each grid box and simulation period, and NorESM driven ensembles have 250 data points, respectively.

### Apparent temperature

The ATG28 index is used as an indicator for heat stress, which is relevant for impacts on human health (Davis et al., 2016). The apparent temperature is computed using the same formulation as in Davis et al. (2016):

$$AT = -2.653 + 0.994T + 0.0153T_d^2, \tag{1}$$

with $AT$ being the apparent temperature, $T$ the daily mean near-surface temperature, and $T_d$ the daily mean near-surface dewpoint temperature. Similar formulations exist in the literature showing very similar results (see Anderson et al., 2013 for a review). The threshold of 28°C is based on the definition of Zhao et al. (2015) who set this limit as the lower boundary for human heat stress.

Future changes of ATG28 are analysed by calculating the differences of the 5th, 50th, and 95th percentiles between current and projected periods. Only grid boxes with 20 or more days over 28°C in the current period were included in the analysis in order to allow for confidence interval calculations for the calculated percentiles using order statistics. Statistical significance is determined when the calculated percentile in the 1.5°C or 2.0°C period is outside the percentile confidence range of the current period. As ATG28 is temperature-based, changes over the ocean surfaces are masked out, because they are determined by the prescribed SST changes to a large extent.

### Five-day precipitation sum

The annual maximum of the five-day precipitation sum (RX5day) is used to characterise heavy precipitation events, which can be relevant for flood generation in river basins. The RX5day represents a noisy, i.e., highly spatially and temporally variable parameter. A large ensemble allows for a better assessment of the signal-to-noise in extreme precipitation.

Differences in RX5day are computed by subtracting the ensemble mean of the current decade from that of the 1.5°C and 2.0°C periods. Statistical significance for RX5day was calculated using a Mann-Whitney-U-test and only results with a significance at the 95% level are shown.

**Daily precipitation intensity**

A change in extreme precipitation directly influences local communities. Such communities have applied design standards for structures to withstand floods with a specified return period. These return standards will no longer be applicable when the extreme value distribution shifts with global warming. A Gumbel Type I extreme value distribution is fitted to the annual maxima of daily rainfall amounts. Using this distribution, an estimate is made of the intensity of rainfall events associated with a given exceedance probability, in line with engineering practices. For each ensemble, the daily rainfall intensity with a 50-year return is computed, hereafter called RI50yr. Information on changes in the rainfall intensity with a 50-year return interval is useful for infrastructure design and maintenance. For example, road authorities in Europe typically use between 1 and 10-year return periods for assessing effects of rainwater falling on major roads (highways), and between 10 and 100 years for rainfall beside the road and waters crossing the road (Bless et al., 2018). Differences in RI50yr between the 1.5°C and 2.0°C simulations compared and the current period simulations are computed as the relative change in mean daily precipitation intensity.

**Consecutive dry days**

Lastly, the Consecutive Dry Days (CDD), defined as the maximum number of consecutive days with a daily precipitation amount of less than 1mm over a region (Karl et al., 1999; Peterson et al., 2001) is calculated for the entire 10-year period of each ensemble member. The CDD is calculated for each of PRUDENCE regions (Christensen and Christensen, 2007), illustrated in Figure 1, because drought indicators are relevant over large areas, as impacts on water resources occur at these scales.

The significance of changes in the CCD distributions is determined using the Mann-Whitney U-Test with a significance at the 95% level. This determines whether samples from the the 1.5°C and 2.0°C simulations are drawn from a population with the same distribution as the current period.

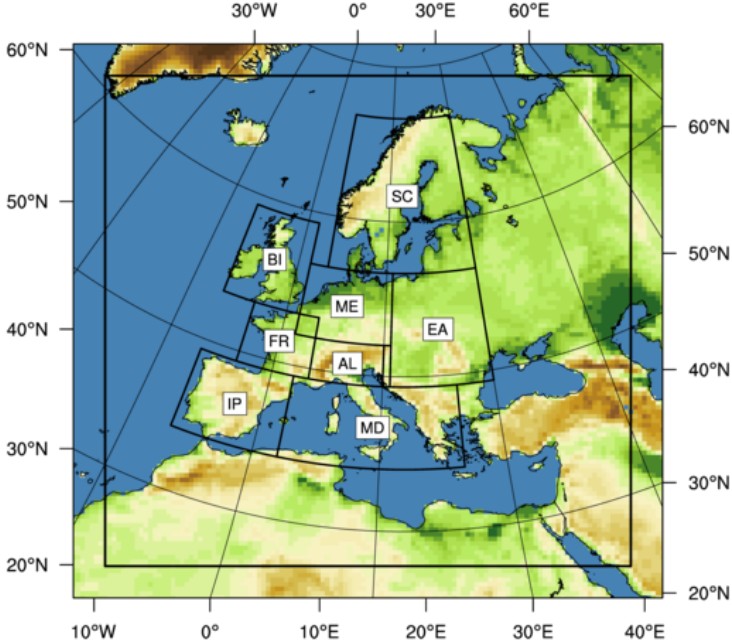

**Figure 1. The European CORDEX domain used by REMO. The outermost black square shows the sponge zone. The PRUDENCE regions are depicted by the boxes with the centred two letter codes: BI=British Iles, IP=Iberian Peninsula, FR=France, ME=Middle Europe, SC=Scandinavia, AL=Alps, MD=Mediterranean, EA=Eastern Europe**

## 3. Results

### 3.1 Apparent temperature

Figures 2 and 3 show the changes in ATG28 for the NorESM and ECHAM6 driven ensembles, respectively. The grey boxes are masked out areas. On land, they refer to grid boxes that do not match our criteria of 20 or more occurrences of ATG28 during the current period. Ocean boxes are masked out, because any change is very closely related to the prescribed SSTs. In general, the changes are strongest close to warm ocean areas, especially around the Mediterranean. But also the central and eastern parts of Europe show increases in ATG28, consistent with the increase in mean temperature (not shown). The distinct difference between the two warming levels should be noticed. Around the Mediterranean the increase in ATG28 during the 1.5° C period is mostly moderate with up to 9 days in the median whereas changes in the 2.0°C period are reaching 18 days and more. This result is consistent between the ECHAM6 and NorESM driven ensembles. For the Northern parts of Spain, at the French Atlantic coast and parts of Eastern Europe, we can note stronger changes for the 95th compared to the 5th percentile. This means that the shape of the distribution changes towards more high extreme values. For most of the other regions there is only little or no change in the shape of the distribution of ATG28. It should be noted that the spatial resolution of the

simulations allows to show the lower level of warming in mountainous areas compared to coastal areas, e.g., over Italy. This is especially important in areas with complex topography such as the Mediterranean region, which is usually only poorly resolved in GCM simulations.

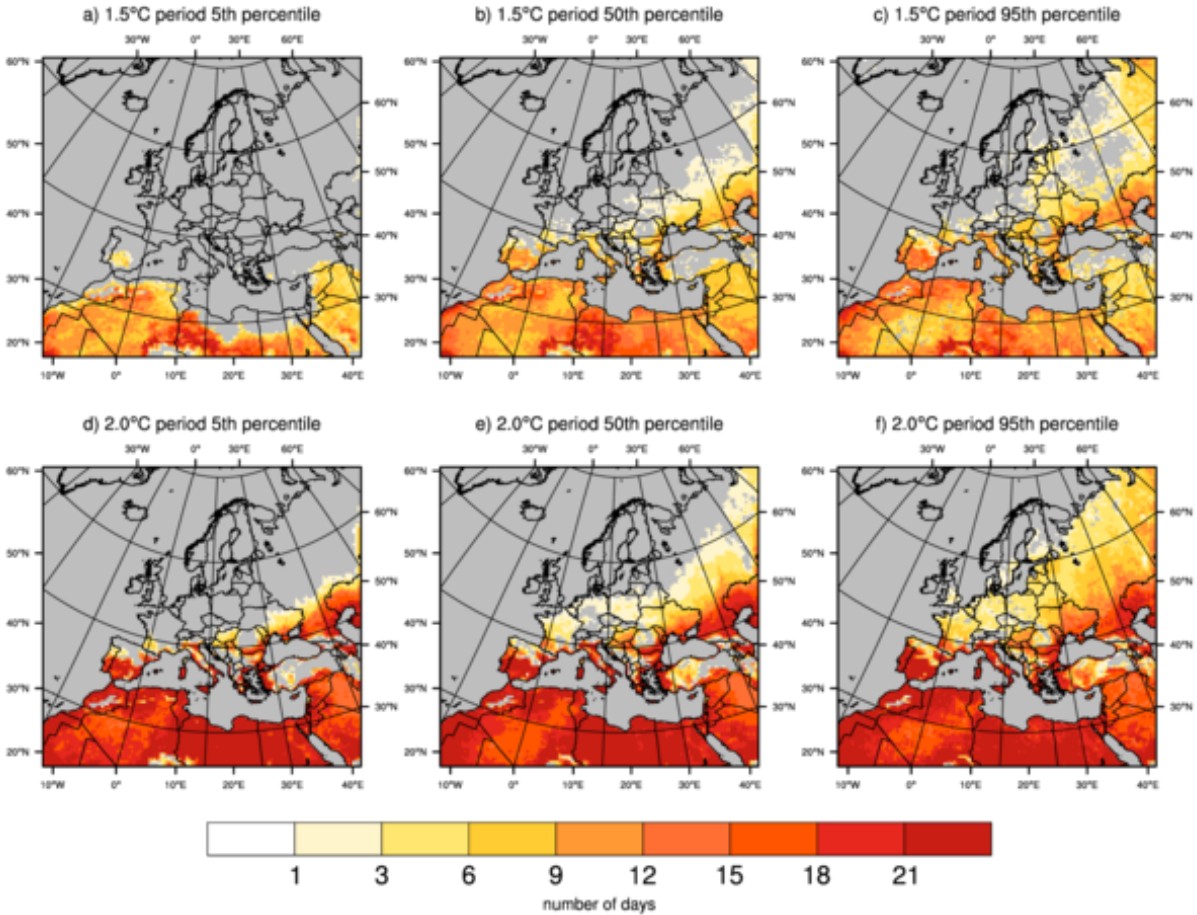

**Figure 2. Differences in ATG28 between the current and the 1.5°C period (top row) respectively the 2.0°C period (bottom row) for the NorESM driven REMO simulations in number-of-days. Shown are the Differences in the 5th percentile (left column), median (middle column) and 95th percentile (right column). Differences over Ocean areas are masked out in grey, as they are closely related to the prescribed SST changes. Masked out areas over land refer to areas with less than 20 occurrences of ATG28 during the current period.**

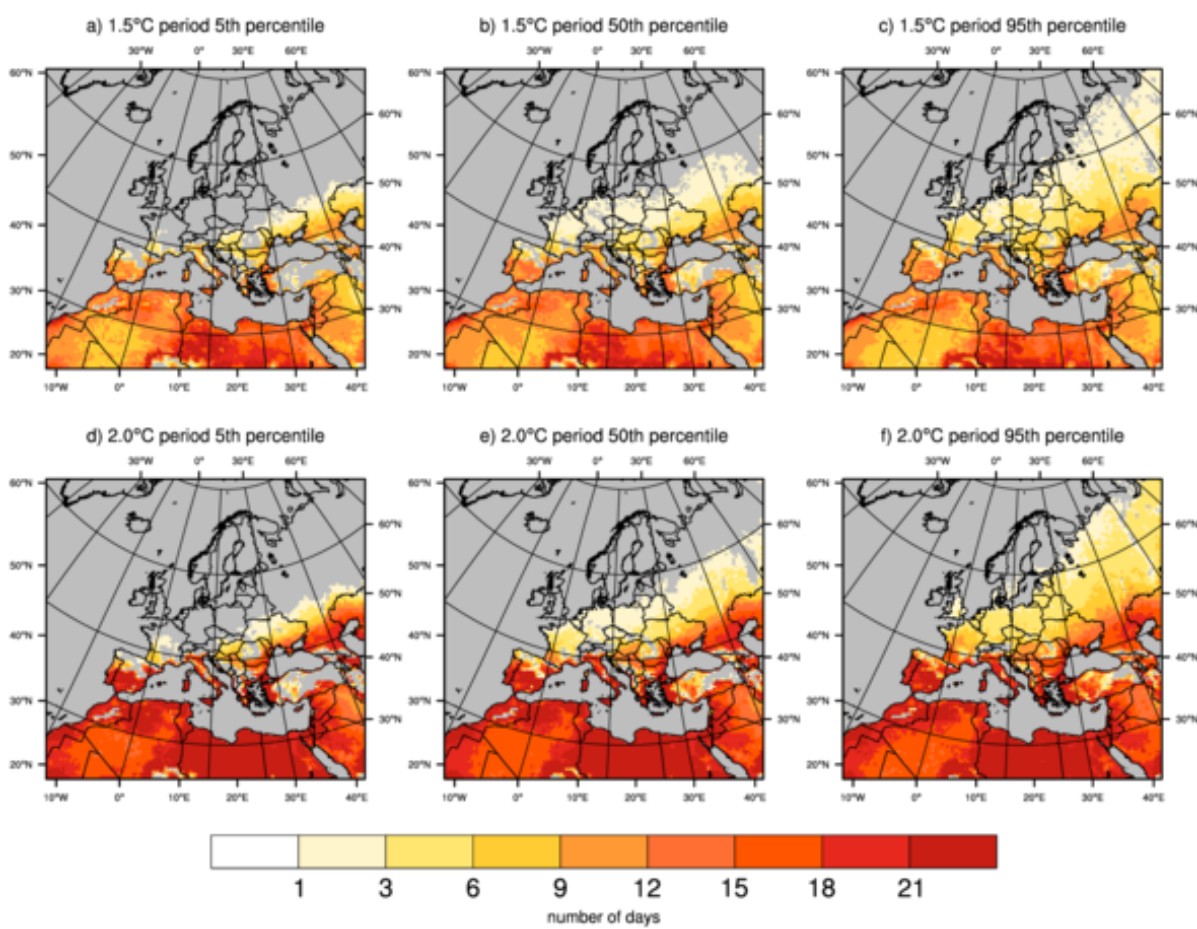

**Figure 3. Same as Figure 2, but for the ECHAM6 driven REMO simulations.**

## 3.2 Five-day precipitation sum

Figure 4 shows the relative differences of RX5day for the REMO ensemble experiments. In general, there is an increase in RX5day over the European part of the domain with stronger signals in the 2.0°C period compared to the 1.5°C period. It can also be seen that the patterns in the ECHAM6 driven ensemble are more coherent than the NorESM ensemble with larger areas showing a significant change. This is related to the difference in ensemble size between ECHAM6 and NorESM driven simulations and underlines the necessity for a large ensemble to achieve proper signal-to-noise ratios when looking at the difference in regional changes under small GMT increases in highly variable quantities such as precipitation extremes. Tests

with a randomly picked 25-member ensemble from the ECHAM6 driven simulations showed a similar noisy pattern as the NorESM driven runs (see Supplement).

Apart from artificial effects due to the boundary conditions, the strongest signal within the core domain appears over the Baltic Sea, with an increase of up to 15% in RX5day under a 2.0°C increase in GMT. This result is consistent between both ensembles. A similar increase can be seen over the Adriatic Sea, but is not so pronounced in the ECHAM6 driven ensemble. This might be related to feedbacks from insufficiently resolved SSTs, because the GCMs usually do not resolve these small sub-basins well. In case of the Adriatic Sea, where precipitation amounts are highly sensitive to changes in local SST (e.g., Stocci and Davolio, 2016), this can lead to biases in heavy precipitation amounts.

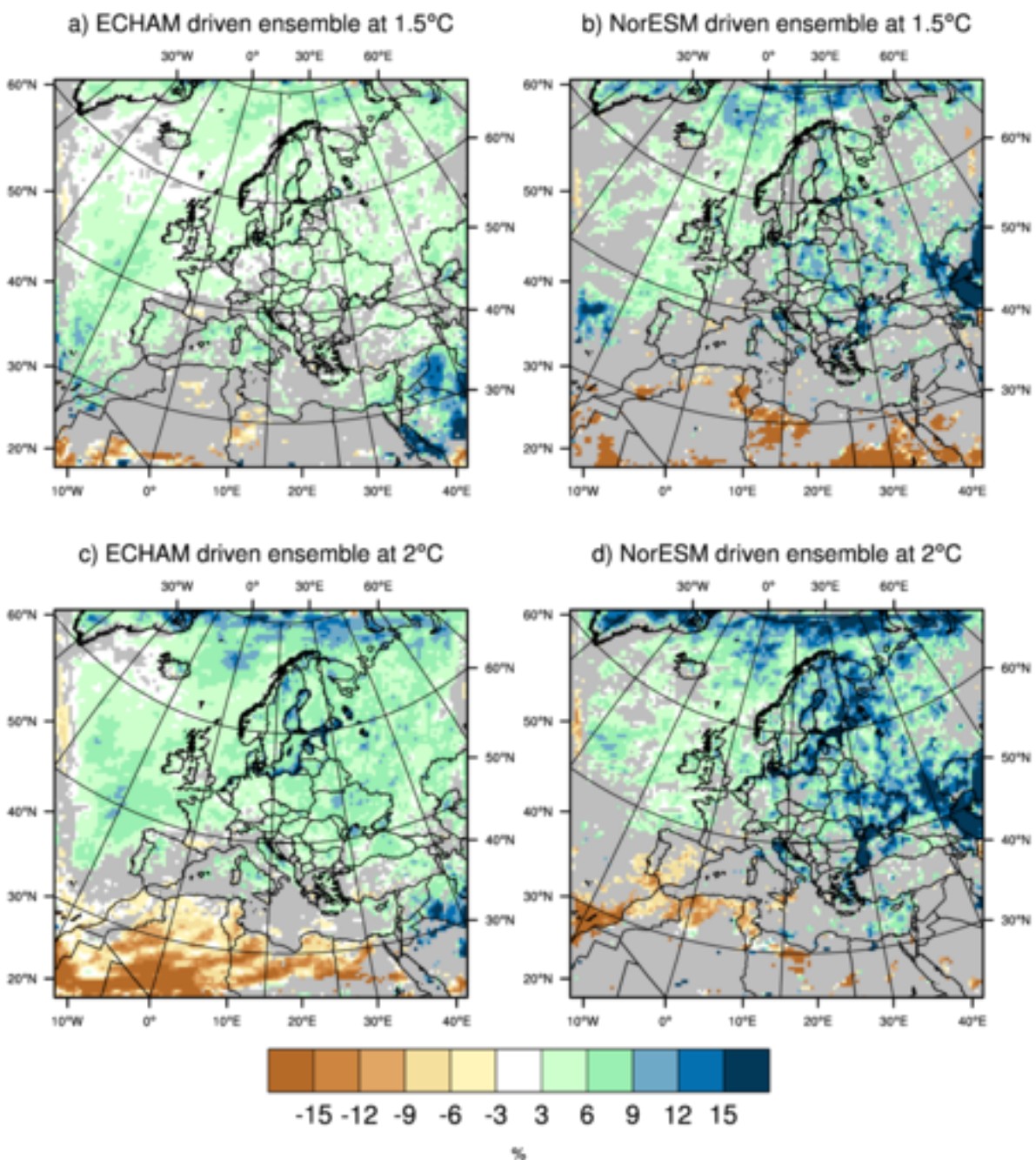

**Figure 4. Relative difference of RX5day (in percent) between current and the 1.5°C period (top row) respectively the 2.0°C period (bottom row) for the ECHAM6 with 100 members (left column) and NorESM with 25 members (right column) driven REMO simulations. Grey shading show areas with non-significant changes on the 95% significance level.**

### 3.3 Daily rainfall intensity, 50-year return period

Figure 5 shows spatial differences in the 50-year return period of daily rainfall intensity (RI50yr) across Europe. In both the NorESM and ECHAM6 driven ensembles, a greater increase in the rainfall intensity is found in the 2.0°C simulations compared to 1.5°C. ECHAM6 driven simulations clearly show increases in RI50yr of up to 20% over continental Europe. The estimated changes in rainfall intensity in the NorESM driven simulations appear to be more extreme but these simulations are also more noisy as they are based on fewer ensemble members.

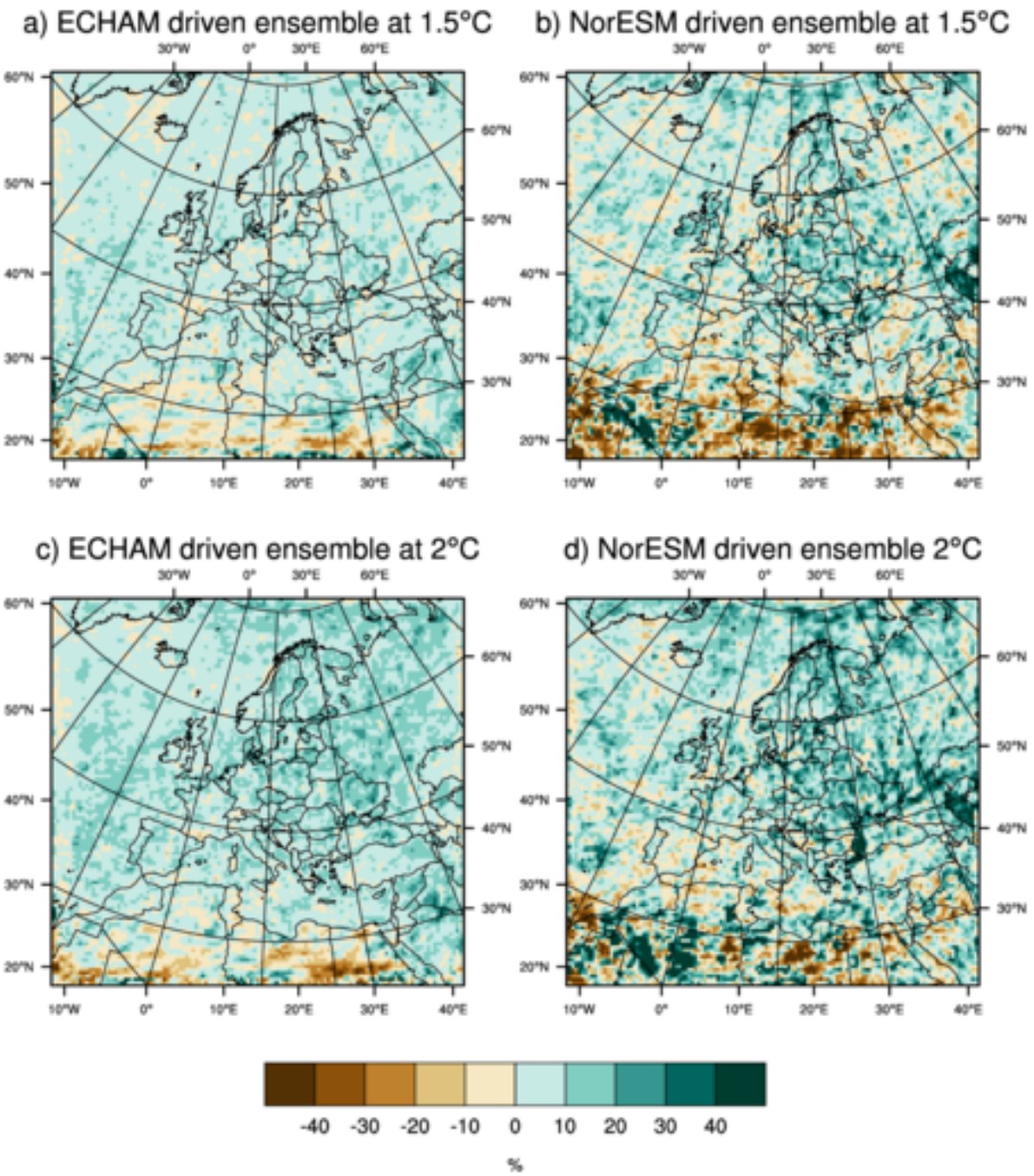

**Figure 5. Relative difference of RI50yr (in percent) between current and the 1.5°C period (top row) respectively the 2.0°C period (bottom row) for ECHAM6 with 100 members (left column) and the NorESM with 25 members (right column) driven REMO simulations.**

## 3.4 Consecutive dry days

In this section, changes in the distributions of Consecutive Dry Days (CDD) for the 1.5°C and 2.0°C periods compared to the current period are presented. To distinguish whether these changes in the distributions are statistically significant we employ the Mann-Whitney U-Test. When the resulting p-values of the test are less or equal to 0.05, the null hypothesis is rejected indicating the distributions differ. The p-values for each of the PRUDENCE regions (Christensen and Christensen, 2007; see Fig. 1) are presented in Table 1. Bold numbers in Table 1 indicate that the distributions are significantly different.

**Table 1. Mann-Whitney U-Test p-values for distributions of Consecutive Dry Days (CDD) for the ECHAM6 and NorESM driven ensembles for different PRUDENCE regions (for locations see Figure 1).**

| PRUDENCE Region: | ECHAM6 | | NorESM | |
|---|---|---|---|---|
| | Current vs 1.5°C | Current vs 2.0°C | Current vs 1.5°C | Current vs 2.0°C |
| 1. British Isles | 0.270 | **0.035** | **0.024** | 0.053 |
| 2. Iberian Peninsula | **0.036** | **0.000** | **0.000** | **0.000** |
| 3. France | 0.391 | 0.230 | **0.015** | **0.002** |
| 4. Middle Europe | 0.077 | **0.015** | 0.363 | 0.212 |
| 5. Scandinavia | 0.238 | 0.356 | **0.046** | 0.081 |
| 6. Alps | 0.333 | 0.105 | 0.378 | 0.355 |
| 7. Mediterranean | 0.069 | **0.036** | 0.348 | **0.021** |
| 8. Eastern Europe | 0.325 | 0.465 | 0.419 | 0.385 |

We begin by looking at three regions where the Mann-Whitney U-Test provided consistent results across the ensembles. In region 2, the Iberian Peninsula, the CDD distributions in both the 1.5°C and 2.0°C simulations differ statistically compared to the simulations for the current period. Over this region, one can see an increase in the duration of the longest dry period and the probability of having a CDD, which exceeds 9 weeks, is greater in the warmer scenarios (Figure 6). In contrast, regions 6 and 8, the Alps and Eastern Europe, have changes in CDD distributions under 1.5°C and 2.0°C that are statistically indistinguishable from the current period (Table 1). One can deduce that region 2, will likely have longer drought periods than experienced before compared to regions 6 and 8. Interestingly, for region 7, the Mediterranean, the CDD distributions of the ECHAM6 and NorESM ensembles of 1.5°C do not differ statistically from the current period, yet both ensembles show a statistically different distribution at 2.0°C. Thus, one can conclude for region 7, according to these simulations, a lower target of 1.5°C increase in GMT could reduce the length of the maximum number of dry days in this region, compared to a 2.0°C target. In region 3, France, the results of the two ensembles differ. The ECHAM6 simulations suggest that there is no difference in CDD distributions between the warmer climate compared to the current period; whereas the NorESM simulations find the warmer climate has a distinctly different CDD distribution compared to the current period.

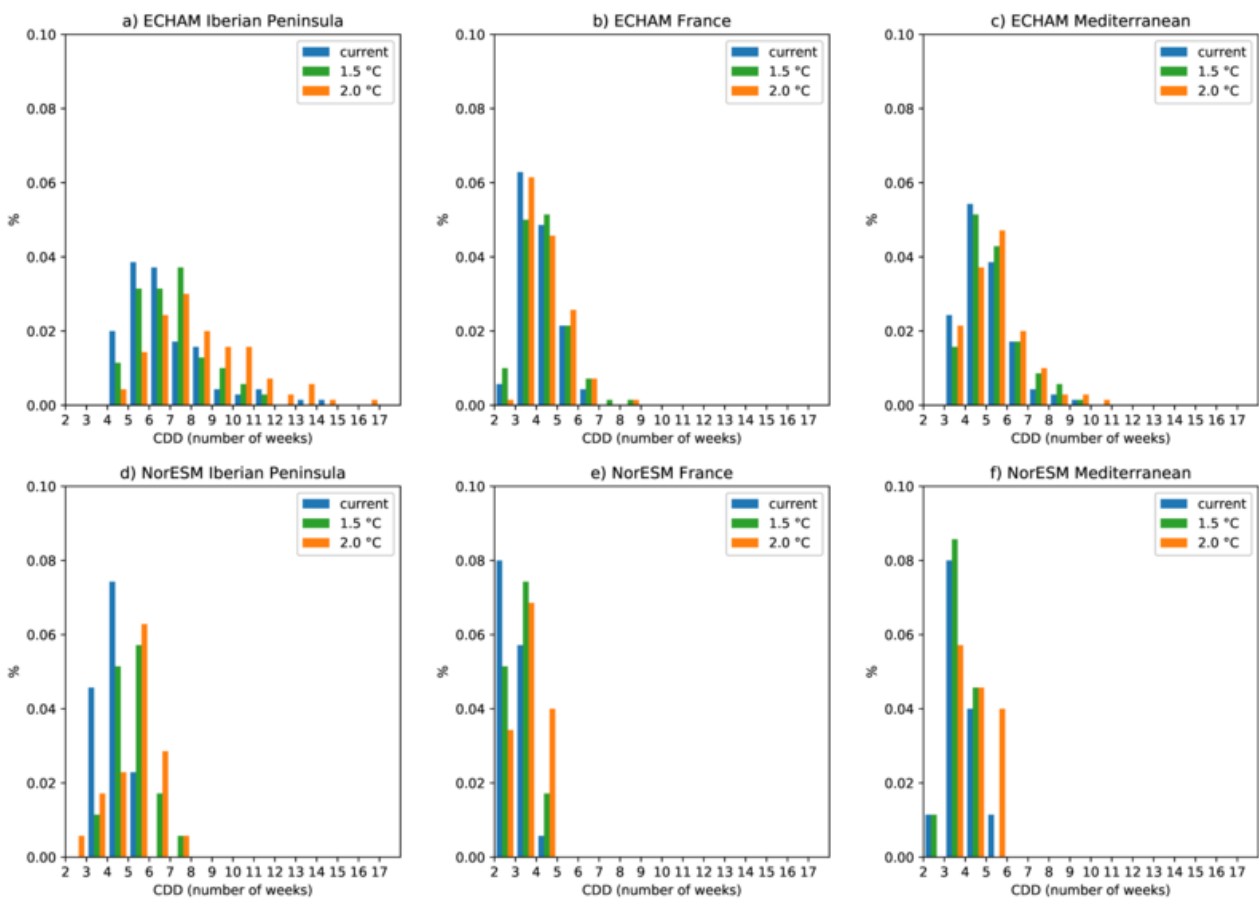

**Figure 6. Duration of drought events in three PRUDENCE regions (2=Iberian Peninsula, 3=France, 7=Mediterranean) under 1.5° and 2.0° global warming for the ECHAM (top row) and NorESM (bottom row) driven ensembles. For significance see Table 1.**

## 4. Discussion and conclusions

A unique climate dataset has been presented that enables the quantification of differences between a 1.5°C and 2.0°C warmer world compared to pre-industrial times on a regional level. This dataset can support climate change impact studies on the regional scale with physically consistent data, which is often not possible to achieve with other methods than dynamical downscaling. The use of a large ensemble (100 x 10 years) compared to alternative datasets for analysing changes under different temperature targets is especially beneficial to assess changes in highly variable meteorological parameters, such as extreme temperature and precipitation.In general, the 100 members driven by ECHAM6 provide information of statistically significant changes over relatively large and spatially homogeneous areas. In comparison, the 25-member ensemble driven by NorESM shows a much noisier spatial pattern which lowers confidence in the projected changes.

The significant differences in apparent temperature ATG28 under different global mean warming level show that a 0.5°C larger global mean warming can have considerable consequences for human health. This is especially true around the Mediterranean, where changes towards more hot and humid conditions along the coasts can have negative impacts on the population and may increase mortality due to heat stress. The tourism sector may also be negatively affected by hotter and more humid conditions. Robust estimates of percentiles and changes in percentiles can be derived from the large ensemble. Some regions show a change of the shape in distribution of ATG28 towards more high extreme values. These findings are not directly comparable to earlier studies, but are in-line with studies on extreme temperatures under 1.5°C and 2.0°C warming (e.g., Schleussner et al., 2016; Sanderson et al., 2017; Suarez-Gutierrez et al., 2018).

RX5day shows a general increase over Europe which is more pronounced under higher global mean warming. More coherent spatial pattern with larger areas showing significant changes result from the larger ensemble driven by ECHAM6 (100 members) compared to the smaller ensemble driven by NorESM (25 members) and also compared to a smaller sub-ensemble driven by ECHAM6, which underlines the need for big ensemble size to reliably detect changes in highly variable quantities such as precipitation extremes. Our results for RX5day are in line with earlier findings by King and Karoly (2017), who investigate RX1day for Europe under 1.5 and 2.0 °C warming. They find that heaviest rainfall events would be more likely under 2 °C warming compared to 1.5 °C.

With regard to the change in the daily rainfall intensity at the 50-year return period (RI50yr), a greater increase in rainfall intensity was found in the 2.0°C warmer world. Given these changes, information can be derived for local communities, which must consider changes in rainfall intensity when designing hydraulic and water resource infrastructures, as well as transportation infrastructure, including highways and bridges. Cost considerations associated with increasing rain intensity demands can be computed for up-coming design projects to ensure investments remain beneficial. The HAPPI dataset can be used to calculate other return periods, catering to the demands of individual sectors.

Robust high and low percentile changes for precipitation are still difficult to distil on a grid box level from the data because of the high variability of precipitation extremes, but methods such as spatial aggregation might help to achieve robust signals on larger spatial scale.

The changes to CDD distributions show that Spain wouldexperience longer droughts in the future compared to the current period, even at a 1.5°C increase in GMT. For Italy, drought conditions associated with the 1.5°C simulations show non-significant changes, yet those associated with the 2.0°C simulations are significantly different to the current period, thus showing possible consequences of exceeding the 1.5°C GMT target of the Paris agreement.

The relatively course resolution of 0.44° for a dynamical downscaling study over Europe is a compromise between model complexity, resolution and ensemble size. In particular, the course driving data of the ECHAM6 model with T63 (1.875°) horizontal resolution does not allow for direct downscaling to a much higher resolution, such as 0.22°, without having a large spatial spin-up of small-scale features inside the domain (see, e.g. Matte et al., 2017). We do not expect fundamental qualitative changes to our findings on a resolution of 0.22°, except where regional mesoscale features play an important role.

As pointed out by Fischer et al. (2018), data from AMIP ensembles such as HAPPI have to be used with caution. In regions and on time-scales where oceanic variability plays an important role, results from AMIP ensembles tend to overestimate extremes and often show overconfidence with a too high significance of the results. For Europe, this effect is especially important on seasonal or longer time-scales. For shorter-term events such as daily extremes, AMIP ensembles compare equally well compared to other methods in the mid- and high-latitudes, because they are dominated by atmospheric variability.

The results for most of the indices presented should not be affected because they specifically target short time-scale extremes. In case of CDD it depends on the region investigated. In some regions CDD comes close to or even exceeds an entire season. In these cases, one should confirm any findings using other sources, e.g., ensembles of coupled GCMs.

The current dataset was created using the only two GCMs available at the time for downscaling, one with a reduced number of ensemble members. As more GCM ensembles become available for downscaling in the future, it will allow for new studies which can provide more robust estimates of inter-model variability/uncertainty. Nevertheless, there is currently a unique dataset targeted to the Paris agreement goals available for further analysis. A comparison with alternative methods for extracting the warming level is lacking and should be done in future studies.

*Data availability*. The datasets generated and analysed for this study are available on request (happi-data@hzg.de). It is planned, to store the data in the long-term archive of DKRZ: http://cera-www.dkrz.de/

*Author contributions*. KS and DJ designed the study. KS performed the regional climate model simulations. KS and CN did the analysis of the climate indices and created the plots. KS, CN, LMB and DR wrote the initial draft paper. All Authors contributed with discussions and revisions.

*Competing interests*. The authors declare that they have no conflicts of interest.

*Acknowledgements*. The authors thank three anonymous referees and Laura Suarez-Gutierrez for providing comments that helped to improve this paper. We also thank the German Climate Computing Center (DKRZ) for supplying computing time and the HAPPI consortium for providing the boundary conditions for the downscaling. We acknowledge the E-OBS dataset from the EU-FP6 project ENSEMBLES (http://ensembles-eu.metoffice.com) and the data providers in the ECA&D project (http://www.ecad.eu). The authors thank their co-workers for the valuable discussions. This work is part of the research project HAPPI-DE supported by the German Federal Ministry of Education and Research (BMBF) under grant number 01LS1613C. It was also supported by the Digital Earth project of the Helmholtz Association, funding code ZT-0025.

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
