# Peer review of "Weather extremes over Europe under 1.5°C and 2.0°C global warming from HAPPI regional climate ensemble simulations"

_Earth System Dynamics, 2020_

## Short Comment (SC1) · 21 Feb 2020

I write this comment because I want to address some aspects that I believe the authors should consider, and would significantly improve the quality and usefulness of their findings.

1. I find there is some lack of context of how the simulations used in this paper compare to those widely used in previous studies. It would be very beneficial to show some type of evaluation of the simulated climate in the two ensembles used here, for example in terms of mean temperature and precipitation anomalies and changes at 1.5 and 2 degrees with respect to preindustrial. Moreover, the authors should describe how the

simulated climates in these ensembles compare to those in previous studies; e.g.., Perkins-Kirkpatrick et al, 2017; Sanderson et al. 2017; King et al. 2017; Suarez-Gutierrez et al., 2018; Wehner et al, 2018. These studies are based on a variety of types of ensembles, from CMIP5 to fully coupled ESM large ensembles, and the paper should include a discussion on how these simulations differ in terms of both climate conditions and fundamental design. In particular highlighting both the advances (i.e. higher resolution, targeted to 1.5 and 2 degrees) and shortcomings (i.e. atm only runs, no fully coupled ocean, SST prescribed from short periods) of these data is in my opinion crucial.

2. The paper does not address the implications of using atmosphere only runs with prescribed SST based on relatively short time periods sufficiently. A finite set of prescribed SST patterns offers a limited range of climate states that does not completely sample ocean-driven variability (see Sanderson et al., 2017; Fischer et al., 2018). In contrast, large ensembles from fully-coupled climate models sample a wider range of ocean states and include the influence of the ocean-borne variability (Hawkins et al., 2016). Furthermore, fully-coupled large ensembles also offer a more realistic representation of heat extremes over land than atmosphere-only large-ensembles, even if the later offer a larger number of independent simulations (Fischer et al., 2018). These issues should be addressed in the main text.

3. The authors argue that the improved resolution from using a regional model combined with the large ensemble size are mayor improvements. However, previous studies analyze the changes at 1.5 and 2 degrees using similarly large ensembles of fully coupled ESM (100 members x 250 years; Suarez-Gutierrez et al., 2018), so what are the differences or biases that higher resolution vs. no coupled ocean introduce?

Minor Comments:

What do the gray areas over land in figures 2, 3, and 4 represent? I thought maybe the white shading was meant to be transparent but there is white in some parts I think?
References:

Fischer et al. (2018): Biased Estimates of Changes in Climate Extremes From Prescribed SST Simulations. Geophysical Research Letters 45.16, pp. 8500–8509. https://doi.org/10.1175/JCLI4288.1

King and Karoly (2017): Climate extremes in Europe at 1.5 and 2 degrees of global warming. Environ. Res. Lett. 12 114031. https://doi.org/10.1088/1748-9326/aa8e2c

Perkins-Kirkpatrick, S.E. and Gibson, P.B. (2017): Changes in regional heatwave characteristics as a function of increasing global temperature. Sci Rep 7, 12256. https://doi.org/10.1038/s41598-017-12520-2

Sanderson et al. (2017): Community climate simulations to assess avoided impacts in 1.5 and 2ŕL'°C futures, Earth Syst. Dynam., 8, 827–847, https://doi.org/10.5194/esd-8-827-2017

Suarez-Gutierrez et al. (2018): Internal variability in European summer temperatures at 1.5L'°C and 2L'°C of global warming, Environ. Res. Lett. 13 064026, https://doi.org/10.1088/1748-9326/aaba58

Wehner et al. (2018): Changes in extremely hot days under stabilized 1.5 and 2.0L'°C global warming scenarios as simulated by the HAPPI multi-model ensemble, Earth Syst. Dynam., 9, 299–311, https://doi.org/10.5194/esd-9-299-2018.

ESDD

---

## Referee Comment (RC1) · Anonymous Referee #1 · 2 Mar 2020

Review comments on Weather extremes over Europe under 1.5°C and 2.0°C global warming from HAPPI regional climate ensemble simulations by Sieck et al. (esd-2020-4)

This paper presents simulations from the HAPPI project, one large and one semi-large global climate model ensemble downscaled by one regional climate model. The results are used to study the climate at +1.5°C and +2°C global warming. This will occur rather soon, which means that climate change is small compared to natural variability. When that is the case there is a clear benefit of using large ensembles since that enables better statistical analyses. This paper could be a good contribution to the

"large ensemble field", especially since it is a RCM ensemble, which is not usually the case. Unfortunately the paper gives a slipshod impression. I wish that the authors would have proofread and language checked once more before submitting. All the questions I get about definitions, methods and analyses stands sometimes in the way for my evaluation of the scientific content. I think the paper needs a lot of improvement before being published.

* General comments

I think Laura Suarez-Gutierrez raises some valid points about how the results presented here relates to previous studies and the impact of the prescribed SSTs (which the authors themselves describe as "unrealistic"). I trust the authors to properly respond to that, so I wont go into that more here.

Is this paper a presentation of a data set or a presentation of results? The title suggests results, but the abstract starts "This paper presents a novel data set" and the Discussion "A unique data set has been presented". It's of course fine to do both, but a data description paper would require a lot more information about models, time periods, scenarios etc. I'm not sure that I agree that a data set is properly "presented" here.

Describe the model experiments in more detail. Why did you use 10 year periods? And why 20 years for the pre-industrial period? How are the specific warmings levels (SWLs) for +1.5 and +2 defined and calculated? You say that you use both RCP2.6 and RCP4.5. Do you mix them in the SWLs? What is the ratio RCP2.6/RCP4.5? I guess that this is described in some HAPPI paper, but it's worth to spend a few lines on that also here.

10 year periods are short in a climatological sense, how is the choice of 10 year periods motivated? One could, of course, argue that with enough ensemble members natural variability will be sampled anyway; however, 10 years with 100 members equals 1000 simulated years which corresponds to 33 members simulating 30 years. 10 years

times 25 members equals 30 years and 8.3 members. A 9 or 35 member ensemble does not sound as impressive as a 25 or 100 member ensemble. Don't make to bold statements about the size of the ensemble. Furthermore, you don't explicitly say how many members the RCM model consists of. You say that the GCM ensembles have 25 and 100 members respectively, but you don't actually say that you downscale all of them. Not as far as I can see at least. Overall I think section 2.1 could be rewritten in a clearer way first presenting the HAPPI project and the GCMs then the RCM and the GCM-RCM combinations, time periods, etc. As it is now it's a bit of a mixture where the reader has to go back and forth to get it all.

Why did you choose these particular indices? And why do you do use different statistical methods to analyse them? The choice of methods and the ways to present the results seem a bit arbitrary.

A suggestion for improving section 2.2: Remove the bullet points with indices. It's a bit strange when the indices are listed together with some kind of motivation or definition, but in a different way for each index. Instead just list the names of the indices. Then, have a sub-heading for each index under which you properly explain the definitions and motivations behind each index.

To what degree would you say that you are showing the value of large ensembles? You mention a smaller sub-ensemble, but I can't see it in the analyses. Sure, you compare the NorESM and ECHAM6 forced ensembles, but how can you know that all differences between the ensembles are due to the ensemble size and not the models themselves?

Think about how you want to name the SWLs. "1.5°C period", "temperature target 1.5°C simulation" (bulky), "2.0°C increase in GMT" or something else. It's a bit annoying when different names are used at different places in the text.

* Specific comments

L20: This sentence (especially the first line) is incomprehensible to me. Think about what you want to say, then write it in two, or even three, sentences. Long sentences with few commas has no intrinsic value.

L26-28: One could add to this that even though it's relevant to work with SWLs the choice of RCP can have an impact on the simulated SWL climate (e.g. Bärring & Strandberg, 2018). This should be interesting to you since I suppose that you mix RCPs in you index calculations.

L30: "5 to 15 models available". I had a quick look at ESGF and found ca 70 members from 30 models from 13 model families. I don't think it's fair to describe the CMIP5 archive in such a diminutive way.

L58: How where the +1.5 +2 SWLs calculated, please explain.

L59: Why is the pre-industrial period 20 years when the other periods are 10 years?

L61: "greenhouse gas forcing is constructed from RCP2.6 and RCP4.5" What do you mean by "constructed"? Don't you just use the forcing data from the RCPs?

L61: "RCP2.6 and RCP4.5" Do you use both and mix them in the SWLs? How many of the 100 (25) members are from RCP2.6 and RCP4.5 respectively?

L69: The use of "per period" is a bit confusing. Isn't it enough to just state the number of models?

L73: "For each GCM member" Are these all of the 125 members?

L74: "These time scales" What times scales?

L75: "RCMs" → "RCM's"

L94: "recommend" → recommended

L97: (or L105-109) Please explain a bit more about apparent temperature. Why is it apparent? Why doesn't it always occur?

L109: "Relative change" I guess you look at the change in all indices, I don't get why you specify this explicitly here.

L 103: "and NorESM has" → "and the NorESM driven have"

L115-116: This sentence is incomprehensible to me. Think about what you want to say, then write that in two, or even three, sentences. Long sentences with few commas has no intrinsic value.

L116: Remove "As such", this is not the correct use of that term.

L118: "exceedance probability" Isn't rainfall events rather associated with either a probability or a threshold. Maybe I just don't understand what you mean.

L118: " rainfall intensity for the 50-year return" Do you mean rainfall intensity with a 50-year return period?

L119: "Such information" What kind of information?

L121: "between 100 and 100 years" I guess that at least on of the "100" should be another number.

L122: What is your definition of CDD? Is it the longest period of consecutive dry days, or something else? Is it the longest period over the whole 10-year period or is the annual average for all 10 years? What is the threshold for a dry day (1 mm?)?

L122: Why do you analyse CDD for the Prudence regions and not in a map as with the other indices? Or, why don't you do the thorough analysis that you do for CDD for the other indices?

L130: "historical" Is this pre-industrial (1861-1880) or current (2006-2015)?

L131-132: "differences of the /.../ percentiles were computed by subtracting the ensemble mean" Isn't the difference in any percentile calculated by computing the difference between the percentile for one period with the same percentile for another period? I

think this sentence could be made more understandable.

L132: "areas" What kind of areas? Isn't it done per grid point?

L135: How is the "percentile confidence range" defined?

L138: Why do you choose the Mann-Whitney-U-test?

L140: "precipitation intensity of the 50-year period" I think you mean "precipitation intensity with a 50-year return time".

L141: "historical" Is this pre-industrial (1861-1880) or current (2006-2015)?

L141: "NOResm" → "NorESM"

L141: Why do you explicitly mention the model names here? I expect you to do analyse both ensembles for all indices. It's implicit that you do.

L150: "mean temperature" Please add a "(not shown)" here.

L152: "and more in the median around the Mediterranean" Please consider rephrasing to something more understandable.

L154: "no change in the distribution of ATG28" Based on figs 2 & 3 I don't agree. For +2 in central Europe the 5th percentile doesn't seem to change much, while the 95th percentile increases with around 6°C. Isn't that a change in the shape of the distribution?

L154:"spatial resolution allows". It's of course better than the GCMs, but is it really true? Isn't the motivation for EUR-011 that EUR-044 doesn't resolve complex topography?

L156"Mediterranean" → "around the Mediterranean" or "Mediterranean region"

Figs 2 & 3. Please consider the following: Add percentile names in a new top row. Add SWL names in a new left column. Add units by the colour bars. Add letters a-f in the caption. Add ensemble sizes in the caption And it seem like white colours are replaced by grey.

L169: "the four REMO ensemble experiments" This is a bit ambiguous. How many REMO ensembles are there, 1, 2, 4, 6? Depends on the definition. Consider erasing "four".

L172: "more coherent" More coherent than what? Not with "larger areas".

L172: "more significant" How is significance calculated, and how is it shown in fig 4?

L173: "difference in ensemble size" Between what?

L176:"the interior of the simulation" What is the "interior of the simulation" if not everything apart from the boundaries? This seems to be an unnecessary complicated way to describe where the largest changes are. Also consider changing "simulation" to "domain".

L179: It should be easy enough to at least roughly test the effect of SST. Just plot it and see how unrealistic it is. Also check the timing of RX5day, is it in winter or in summer? I guess the SST bias works differently in different seasons. In winter it's probably too warm, in summer too cold. I suggest that you do some kind of check.

Figure 4. Consider the following: Add SWL names in a new top row. Add model names in a new left column. Add letters a-d to the panels. Add units to the colour bars. Explain grey shading in caption.

L190: "To account for" What do you mean with this¿It doesn't seem to be the correct use of the term "to account for".

L190: "the relative change in daily rainfall intensity is presented in Figure 5". No, it's not. Figure 5 shows the change in the intensity with a 50 year return time.

L191: "In the both the" → "In both the"

L193: "precipitation intensity of the 50-year period" I think you mean "precipitation intensity with a 50-year return time".

L190-195: It's seems like you're struggling with how to describe the precipitation intensity of events with a 50 year return time. Why don't you just define RI50yr properly and a bit lengthy, and then just stick to RI50yr? That would save you some trouble in writing, and should avoid some confusion for the reader.

Fig 5: Who do you suddenly show results for a different domain? Excluding parts of northern Europe and including parts of northern Africa where you don't have data. For consistency, show the same domain in all plots. This domain should preferably be the same as the model domain, unless you have a good motivation for excluding certain areas. Also, consider the following: Add SWL names in a new left column. Add letters a-d in the caption. Replace "Percent" with "%" in the legend. This is perhaps a matter of taste, but common practice is "%" I think.

L204: This sentence should start: "Both the 1.5°C and 2.0°C distributions ..."

L205: "historical" Is this pre-industrial (1861-1880) or current (2006-2015)?

L205: I think you can remove "respectively". It doesn't add anything.

L207-208: "whereas the /.../ distributions" This goes without saying. Consider removing for brevity.

Table 1: Why are the p-values suddenly the most important part of the analysis of an index? And why is CDD analysed for the Prudence regions? Please explain.

L215: "longer period of dry days" Be careful how you interpret changes in CDD. You don't define CDD so I can't be sure if you make the correct interpretation. If your CDD is averaged over your 10- year period it could be correct. If your CDD is the longest dry period over the 10-year period it only tells you that the longest dry period will be longer. That doesn't necessarily mean that dry periods on average will be longer.

L217: ". . . indistinguishable in the simulations /.../ (Table 1)." I don't understand the this sentence. Please rewrite.

L218: "more frequent" This is not correct. CDD is the length of the longest dry period (at least this is the common definition). If you want to know if dry periods will be more frequent you should study the number of dry periods.

L221: "1.5°C vs. 2.0°C" I think yo mean "1.5°C instead of 2.0°C"

L222: "adaption" Do you mean "adaptation"?

L221-222: I have a few problems with this, somewhat ambiguous, sentence. First, it is not the +1.5°C or +2°C targets that will have an impact on society, it is the climate change. Second, that the climate will be different in the +1.5°C world compared to the +2°C world is obvious. I guess you mean that the change in CDD is not linear so that the extra 0.5°C will have a large impact. Third, why do you point out changes in CDD in region 7 as a particular motivation for adaptation and mitigation? In my view this whole paper is a motivation for adaptation and mitigation as it shows how climate change may change in the future. Consider rewriting.

L225-226: It's a poor motivation to exclude regions just because the U-test gives different results. Especially since the results differ also in region 3. Strictly speaking the results differ for all regions since you get different p-values (Table 1). With the possible exception of IP at +2.0 where both ensembles get 0.000.

Fig 6: It's very odd to measure the number of days in the unit weeks. Add to the caption something like: "for the ECHAM (top row) and NorESM (bottom row) driven ensembles".

L235: "10 x 100 years" I would prefer "100 x 10 years" since it is 100 10-year simulations.

L250: "smaller sub-ensemble driven by ECHAM6" I don't see this sub-ensemble anywhere in the text. Should it be added to the analysis?

\* References

Bärring, L., & Strandberg, G. (2018). Does the projected pathway to global warming targets matter? Environmental Research Letters, 13(2), 024029. https://doi.org/10.1088/1748-9326/aa9f72
* * *

---

## Referee Comment (RC2) · Anonymous Referee #2 · 24 Mar 2020

The paper investigates the impacts of 1.5°C and 2.0°C global warming on temperature and precipitation extremes over Europe. With this aim, the authors use an ensemble of dynamically downscaled simulations from the HAPPI project. The analysis focusses on four climate indices from ETCCDI.

The paper covers an interesting and relevant topic that could be a useful addition to the literature. Unfortunately, several aspects in terms of methods, analyses and results are unclear and need further explanation. In fact, the manuscript needs a lot of improvement in order to increase the clarity and readability. Please refer to the main points and specific comments below. In particular, the authors do not provide enough convincing

evidence to prove the benefits of their method. Nevertheless, I suppose that there are sufficient arguments for it. Therefore, I suggest a major revision for this manuscript.

MAIN POINTS

1) Text: The whole manuscript needs a thorough proofreading and language check. Some sentences are incomprehensible; others are just too long and should be split in two to enhance readability. In addition, several phrases/descriptions are not consistent throughout the paper and therefore may confuse the reader. Please also refer to specific comments below.

2) Figures: The figures and their captions need some general improvement.

2a) Why are some land areas in Figures 2-4 grey?

2b) You should use the same format for all spatial plots, e.g. * One colour bar including units * Label individual plots * Same domain * Add more information in the figure itself (warming level, ensemble, . . .).

2c) Figure 6: The bars are difficult to distinguish. Non-overlapping bars might be preferable. The distributions for both GCMs are very different. For NorESM, they are quite narrow, while they are much broader for ECHAM6. What are the implications of such large discrepancies? Please discuss. Why do you measure consecutive dry days in weeks?

3) Is the main objective of this paper to present a new data set or to present mainly new results? Either way, both are not properly presented and need more details.

4) Missing details/explanations: Some points (especially in the Methods section) need a better/more detailed explanation to be understandable. Your descriptions are too short and raise more questions than they answer.

4a) L58/59: Why did you use 20 years for the pre-industrial period, but only 10 years for all other simulations? From a climatological perspective, 10 years are rather short
to enable a climatological view.

4b) L58: Which period did you use for the future climate simulations? There are a few references given, but it should be specifically described which methods have been used here and how they are applied.

4c) Did you compare the simulation for a current decade to observations and/or reanalyses to check how realistic they are? This would be very important.

4d) L59-61: How are the warming levels (1.5 and 2.0) calculated? Did you use RCP2.6 for 1.5° warming and RCP4.5 for 2.0°C warming?

4e) L73: The regional ensemble consists of 125 members, correct?

4f) L122-124: How exactly did you define CDD? Why did you calculate the CDD for the PRUDENCE regions and not on grid-point basis as the other indices? Is CDD the maximum number per year or over the 10-year period?

4g) L130-145: Why did you choose different statistical methods to investigate the individual climate indices?

4h) L130-145: The application of the significance measures is unclear, please rewrite. There are two methods used for two different parameters (ATG28 and RX5day). There seems to be some confusion on what is used for CDD (L143 says CDD similar to ATG28, but the method for CDD seems to be similar to RX5day, namely Mann-Whitney). No information is given for RI50yr. Anyway, the paper provides only information on the significant changes for CDD, but not for the other parameters. This should be remedied.

4i) Why do you think that all differences between the two ensembles are due to the different ensemble sizes (e.g. L172-175)? They could also result from the driving GCMs. It might be useful to include the results for the smaller sub-sample of ECHAM6 that you mentioned in the text.
4j) Table 1: Partly, the smaller ensemble (NorESM) generates more significant results. How does this relate to the hypothesis that a larger ensemble size is beneficial?

4k) Compare your results to previous studies (see interactive comment by Laura Suarez-Gutierrez for more details).

4l) The authors should not oversell their results (or should argue more convincingly). E.g. Impact of the ATG28 increase (L241-246): What does such a change really mean w.r.t health issues? I assume that the number of days above the threshold is already high around the Mediterranean? Does a change of O(10days) drastically change the base level and/or the potential health impacts in this region? Furthermore, is a resolution of 50km sufficient to derive change estimates for local adaptation measures? The authors do not provide enough convincing evidence for the benefits of their method. Nevertheless, I suppose that there are sufficient arguments for it, but this should be better phrased.

SPECIFIC COMMENTS

Data set, dataset or data-set?

NOResm or NorESM?

L17: "measures at a" – Delete "a".

L20-22: This sentence is incomprehensible, especially the first part.

L25: "current generation global climate simulations" –> "current generation of global climate simulations"

L45-47: Please rephrase sentence.

L55-56: You cannot downscale an RCM using GCM simulations.

L59: What does "CMIP5 mean SST anomaly" actually mean? All CMIP5 models and members, or just some, only ECHAM6/NorESM? Is there a reference?
L66/67: "from the core domain defined by CORDEX the entire domain has 121x129 grid boxes" – Please reword.

L94: "recommend" –> "recommended"

L97-100: You defined an abbreviation for each climate index. Use them more consequently throughout the text.

L99: "precipitation intensity at the 50-yr return period" – Do you mean precipitation intensity with a 50-yr return period? Please use a consistent explanation throughout the text.

L111: What do you mean with "annual sum maximum"?

L121: "between 100 and 100 years" – Please correct.

L130: What do you mean with "historical"? Pre-industrial or current? Same in L141, L205, L215.

L132: "Only areas with more than 20 non-zero data points" – Isn't the calculation done at every single grid point?

L152: "in the median around the Mediterranean" – Please reword.

L154: Is a spatial resolution of 0.44° high enough to resolve complex topography? What about EUR-11?

L176: "interior of the simulation domain" – Please reword.

L190-191: Please reword sentence.

L204: You never defined/explained p-values.

L204: "Both the distributions 1.5°C and 2.0°C" – Please reword.

L215: "shift towards longer periods of dry days" – This depends on your definition of CDD. If CDD is the maximum number of consecutive dry days, your results only show

that the longest dry period is getting longer in a warmer climate. That does not mean that all dry periods will be longer.

L216: "the Alps and Eastern European region" –> "the Alps and Eastern Europe"

L218: "suffer from more frequent and longer drought periods" – Again, this depends on your definition of CDD. If you used the common one (CDD being the length of the longest dry period), you cannot say anything about the frequency of dry periods.

L222: "adaption" should be "adaptation"

Figure 6: Use ECHAM6 instead of ECHAM.

L235: "10 x 100 years" –> "100 x 10 years"

L247: "RX5day" –> "(RX5day)"

L251: "such precipitation extremes" –> "such as precipitation extremes"

L263: "historically similar" – Strange wording.

L263-264: "pre-industrial period" – I thought you were calculating differences between future and current climate and not between future and pre-industrial?

―――――――――――――――

---

## Referee Comment (RC3) · Anonymous Referee #3 · 31 Mar 2020

Comments on: Weather extremes over Europe under 1.5◦C and 2.0◦C global warming from HAPPI regional climate ensemble simulations by Sieck et al. (esd-2020-4)

This paper presents results from a regional dynamical downscaling of two GCM ensemble from the HAPPI project and compares the model output to 4 extreme event indices.

The large number of ensemble members and important can potentially make a good contribution in how to analyse and estimate the difference between relatively close climate change "targets".

One major discrepancy however is that there are no evaluation of the quality of results with respect to observations or reanalysis. If the model results deviates to much from the "real world" one can not trust the conclusions given for the warm periods. The discussion does not have to be very advanced but I think you should include a point on this.

In general both the description of methods and results are not always very clearly defined. Sometimes it is just a matter of language. The sentences are sometimes so long that the reader loose the thread before getting to the end, so please try to be more focused.

A problem is that it is sometimes unclear whether the assumptions used are your / RCM limitations or inherited from the Happi protocol. Even though I know the Happi protocol it was sometimes hard to separate. I see that many of the questions from other reviewers on critical assumptions e.g. how does pre-industrial come into play, when is the future time slice .. is often related to the protocol. I think the authors should include a summary of this information, in particular as you rightly point out that the Happi protocol is quite different from the traditional CMIP settings.

The defintions/ presentations of the indices and how they are used can be improved (Section 2.2 and 2.3)  It is particularly hard to follow the set-up of significance tests.
"RX5 days are computed ...similar to ATG28, however …

CDD is similar to ATG28 but with the method used for RX5 but now however?
There are no significance test for RI50yr ? There does not have to be one but I would like to know.

With only 4 indices I think you should avoid "similar" as much as possible and just describe the methods for each.

The discussion on being able to reproduce the more noisy results of the smaller ensemble by picking a smaller number of the largest ensemble was interesting. I wonder if it is any way of presenting this in a cumulative manner, with respect to number of ensemble members. The change rate may also indicate whether even the largest ensemble is too small.  I realize however that this is likely beyond the scope of the article.

To the point on data availability: Is it the data used specifically for this study, or is it general RCM output.

Some minor suggestions /corrections

Line 20: "differ" not needed
Line 25: Include a line on scenario definition in CMIP6?
Line 33:skip the word "indeed"
line 70: Green-house-gas -→greenhouse gas
line 81: Sea-Ice → sea-ice  (usually not capital letters)
line 121: Missing intervall ("100" years on both sides)
Figure 1. Is the outer map equal to the model domain?

Line 135-136 The explanation is fine, but then you do not need to mask it out either. It would show up as insignificant?

Line 141 NOResm → NorESM

Figure 6. The difference between the two downscaling sets are quite large. Any comments.

line 260 -265 use significance   instead of unlike / similar

---

## Author Comment (AC2) · 2 Jun 2020

**Answer to SC#1**

General response:

The authors would like to thank Dr. Suarez-Gutierrez for the time and effort put into this review. These comments have been useful in improving our manuscript. We have carefully read the comments and provide a detailed response to each comment.

I write this comment because I want to address some aspects that I believe the authors should consider, and would significantly improve the quality and usefulness of their findings.

1. I find there is some lack of context of how the simulations used in this paper compare to those widely used in previous studies. It would be very beneficial to show some type of evaluation of the simulated climate in the two ensembles used here, for example in terms of mean temperature and precipitation anomalies and changes at 1.5 and 2 degrees with respect to preindustrial. Moreover, the authors should describe how the simulated climates in these ensembles compare to those in previous studies; e.g.., Perkins-Kirkpatrick et al, 2017; Sanderson et al. 2017; King et al. 2017; Suarez-Gutierrez et al., 2018; Wehner et al, 2018. These studies are based on a variety of types of ensembles, from CMIP5 to fully coupled ESM large ensembles, and the paper should include a discussion on how these simulations differ in terms of both climate conditions and fundamental design. In particular highlighting both the advances (i.e. higher resolution, targeted to 1.5 and 2 degrees) and shortcomings (i.e. atm only runs, no fully coupled ocean, SST prescribed from short periods) of these data is in my opinion crucial.

Answer: We agree that the inclusion of an evaluation will improve the paper and put the results into context. We will add a discussion of the model chain performance as a supplement to the paper. This will also allow us to put our results into perspective compared to results from other studies. We think a full discussion on the benefits of coupled vs. uncoupled GCM simulations would be beyond the scope of this paper. In this context, we see ourselves as users of a widely discussed dataset. We compare ourselves to other downscaling activities such as EUROCORDEX.

2. The paper does not address the implications of using atmosphere only runs with prescribed SST based on relatively short time periods sufficiently. A finite set of prescribed SST patterns offers a limited range of climate states that does not completely sample ocean-driven variability (see Sanderson et al., 2017; Fischer et al., 2018). In contrast, large ensembles from fully-coupled climate models sample a wider range of ocean states and include the influence of the ocean-borne variability (Hawkins et al., 2016). Furthermore, fully-coupled large ensembles also offer a more realistic representation of heat extremes over land than atmosphere-only large-ensembles, even if the later offer a larger number of independent simulations (Fischer et al., 2018). These issues should be addressed in the main text.

A: We agree that we should mention these shortcomings and we will include this. A full discussion on coupled vs. uncoupled GCM simulations would be beyond the scope of our paper (see answer above).

3. The authors argue that the improved resolution from using a regional model combined with the large ensemble size are mayor improvements. However, previous studies analyze the changes at 1.5 and 2 degrees using similarly large ensembles of fully coupled ESM (100 members x 250 years; Suarez-Gutierrez et al., 2018), so what are the differences or biases that higher resolution vs. no coupled ocean introduce?

A: We argue that extremes can be better estimated using our downscaled information, which has been shown several times in many studies, especially for precipitation. As mentioned earlier, our focus is on downscaling an existing dataset (in this case HAPPI) with a well-established method (dynamical downscaling with a regional climate model). We think the question raised here, is very much related to the comments above and beyond the scope of our paper.

What do the gray areas over land in figures 2, 3, and 4 represent? I thought maybe the white shading was meant to be transparent but there is white in some parts I think?

A: The grey areas over land represent non-significant changes. This was mentioned in the text and will be added to the figure captions (see also our response to referee#2).

---

## Author Comment (AC3) · 2 Jun 2020

**Answers to Referee#1**

General response:

The authors would like to thank the referee for the time and effort put into this review. These comments have been useful in improving our manuscript. We have carefully read the comments and provide a detailed response to each comment.

I think Laura Suarez-Gutierrez raises some valid points about how the results presented here relates to previous studies and the impact of the prescribed SSTs (which the authors themselves describe as "unrealistic"). I trust the authors to properly respond to that, so I wont go into that more here.

Answer: We have responded to each of Laura Suarz-Gutierrez's comments and will revise the text to indicate how our results compare to earlier studies. Please see our responses to her comments. We did not describe the prescribed SST as "unrealistic". In fact, for the current period they are even based on observations. Due to an interpolation issue with the anomalies used in the future period and a special treatment of sea ice at a given SST in the ECAHM model, unrealistic SST jumps occurred in a very few grid boxes. This was what we were referring to and we hope that the new section 2 (see below) makes this now more clear.

Is this paper a presentation of a data set or a presentation of results? The title suggests results, but the abstract starts "This paper presents a novel data set" and the Discussion "A unique data set has been presented". It's of course fine to do both, but a data description paper would require a lot more information about models, time periods, scenarios etc. I'm not sure that I agree that a data set is properly "presented" here.

A: The intention of this paper is to present a new dataset and show some examples of its applications. We propose to extensively rewrite the Methods section (please refer to response to reviewers next two questions, below) to provide a more extensive description of the model experiments, as the referee suggests.

Describe the model experiments in more detail. Why did you use 10 year periods? And why 20 years for the pre-industrial period? How are the specific warmings levels (SWLs) for +1.5 and +2 defined and calculated? You say that you use both RCP2.6 and RCP4.5. Do you mix them in the SWLs? What is the ratio RCP2.6/RCP4.5? I guess that this is described in some HAPPI paper, but it's worth to spend a few lines on that also here.

A: Indeed, the experiment specification is described in other papers from the HAPPI community. We will update section 2 (see our response to the next comment, below) to better explain the HAPPI protocol.

10 year periods are short in a climatological sense, how is the choice of 10 year periods motivated? One could, of course, argue that with enough ensemble members natural variability will be sampled anyway; however, 10 years with 100 members equals 1000 simulated years which corresponds to 33 members simulating 30 years. 10 years C2 times 25 members equals 30 years and 8.3 members. A 9 or 35 member ensemble does not sound as impressive as a 25 or 100 member ensemble. Don't make to bold statements about the size of the ensemble. Furthermore, you don't explicitly say how many members the RCM model consists of. You say that the GCM ensembles have 25 and 100 members respectively, but you don't actually say that you downscale all of them. Not as far as I can see at least. Overall I think section 2.1 could be rewritten in a clearer way first presenting the HAPPI project

and the GCMs then the RCM and the GCM-RCM combinations, time periods, etc. As it is now it's a bit of a mixture where the reader has to go back and forth to get it all.

A: We will address the choices made in the HAPPI project in a dedicated sub-section, as all referees had questions on the experiment set-up as described in our paper. Therefore, we will restructure the Methods section (2.) in the following manner:

"2 Methods

[revised manuscript text omitted]

The rest of section 2 in the manuscript will be kept, but section 2.2 will become section 2.3 (except for changes responding to the specific comments).

Lierhammer, L., Mauritsen, T., Legutke, S., Esch, M., Wieners,K.-H., and Saeed, F.: Simulations of HAPPI (Half a degree Additional warming, Prognosis and Projected Impacts) Tier-1 experiments based on the ECHAM6.3 atmospheric model of the Max Planck Institute for Meteorology (MPI-M), http://cera-www.dkrz.de/WDCC/ui/Compact.jsp?acronym=HAPPI-MIP-global-ECHAM6.3, 2017.

He, J. and Soden, B. J.: The Impact of SST Biases on Projections of Anthropogenic Climate Change: A Greater Role for Atmosphere-only Models?, Geophys. Res. Lett., 43, 7745–7750, 2016.

Why did you choose these particular indices? And why do you do use different statistical methods to analyse them? The choice of methods and the ways to present the results seem a bit arbitrary.

A: Our intention was to show example applications for the regional HAPPI dataset. Two main aspects of the dataset are the high resolution and the large number of ensemble simulations. These aspects allow one to focus on other indices beyond mean annual warming. We have chosen to cover indices related to extreme temperatures and precipitation. These indices carry direct relevance for several applications in health and water management, two sectors directly affected by weather, and thus these indices are not arbitrary. The presented indices are highly accepted indices for extremes. The choice of the indicators are supported by the mentioned reference from WMO and others. Obviously, many other indices can be chosen, and the dataset also allows further analysis by third parties.

A suggestion for improving section 2.2: Remove the bullet points with indices. It's a bit strange when the indices are listed together with some kind of motivation or definition, but in a different way for each index. Instead just list the names of the indices. Then, have a sub-heading for each index under which you properly explain the definitions and motivations behind each index.

A: We will adjust the text, and remove the bullets and add separate headings for each indicator.

To what degree would you say that you are showing the value of large ensembles? You mention a smaller sub-ensemble, but I can't see it in the analyses. Sure, you compare the NorESM and ECHAM6 forced ensembles, but how can you know that all differences between the ensembles are due to the ensemble size and not the models themselves?

A: The value of the large ensemble is clearly given when looking at precipitation. In the manuscript, we compare 25 NorESM members to 100 ECHAM6 members. We find more robust results in terms of spatial patterns when using the 100 ECHAM6 driven members. An analysis of a sub-sample of 25 ECHAM6 driven members confirms this and we will add this analysis as a supplement. Of course there will still be a difference between ECHAM6 and NorESM, which are then model related.

Think about how you want to name the SWLs. "1.5 C period", "temperature target 1.5 C simulation" (bulky), "2.0 C increase in GMT" or something else. It's a bit annoying when different names are used at different places in the text.

A: We agree that our style to refer to the periods is misleading. We will harmonize this and change it to "current period", "1.5°C period" and "2.0°C period".

SPECIFIC COMMENTS

L20: This sentence (especially the first line) is incomprehensible to me. Think about what you want to say, then write it in two, or even three, sentences. Long sentences with few commas has no intrinsic value.

A: We will rephrase the sentence as follows:

"Identifying regional climate change impacts for different global mean temperature targets is increasingly relevant to both the private and public sectors. In the private sector, investors demand financial disclosure associated with climate change risks and opportunities (Goldstein, et al., 2018). In the public sector, policy makers rely on climate information build on internationally agreed limits to develop national climate action policies."

L26-28: One could add to this that even though it's relevant to work with SWLs the choice of RCP can have an impact on the simulated SWL climate (e.g. Bärring & Strandberg, 2018). This should be interesting to you since I suppose that you mix RCPs in you index calculations.

A: Given that the HAPPI protocol is different from time periods associated with RCPs by construction, this should not be relevant to our indices. We will add a sub-section on HAPPI which should make this clear.

L30: "5 to 15 models available". I had a quick look at ESGF and found ca 70 members from 30 models from 13 model families. I don't think it's fair to describe the CMIP5 archive in such a diminutive way.

A: We are only referring to the cited studies. These studies used 5 to 15 models. This is not a general statement on the number of simulations available on ESGF at that time. We hope this re-formulation will make it more clear:

"These studies typically used the 5 to 15 ensemble members which were available in CMIP5 at the time for their global and regional studies."

L58: How where the +1.5 +2 SWLs calculated, please explain.

A: This will be clarified in an additional sub-section on the HAPPI method.

L59: Why is the pre-industrial period 20 years when the other periods are 10 years?

A: The pre-industrial period is only used as a baseline to define the period with 0°C warming. The definition is coming from the HAPPI protocol that is followed by every group doing global simulations. We are aware that there are several slightly different definitions of this particular period. The IPCC Special Report on 1.5°C lists several of these definitions. We will add a new sub-section on the HAPPI experiments to the manuscript (see previous answerers) to make these definitions more clear. Also, the period of ten years comes from the HAPPI protocol and has been discussed by Mitchell et al. (2017). We will add the motivation for the ten year period to the HAPPI sub-section.

L61: "greenhouse gas forcing is constructed from RCP2.6 and RCP4.5" What do you mean by "constructed"? Don't you just use the forcing data from the RCPs?

A: This is again related to the HAPPI protocol. We are not using boundary forcing from classical RCP driven global model simulations, but AMIP style GCM simulations following the HAPPI protocol. The sub-section on HAPPI will make this clearer.

L61: "RCP2.6 and RCP4.5" Do you use both and mix them in the SWLs? How many of the 100 (25) members are from RCP2.6 and RCP4.5 respectively?

A: The dataset is not from mixed RCP2.6 and RCP4.5 simulations. See answer above.

L69: The use of "per period" is a bit confusing. Isn't it enough to just state the number of models?

A: "per period" refers to the three simulation periods (current, 1.5°C, 2.0°C) following the HAPPI protocol mentioned in line 58. Either one has to write 300 and 75 members or the number of members per period. That latter seemed more intuitive to us, especially when we do analysis on simulated slices instead of continues simulations.

L73: "For each GCM member" Are these all of the 125 members?

A: This refers to all 375 simulations done with REMO. We did not construct a regional ensemble using only one GCM member.

L74: "These time scales" What times scales?

A: On a time scale of 10 years. We will change the text to:

"For each GCM member only one REMO simulation was carried out, as inter-member variability of an RCM ensemble over Europe on a time scale of 10 years is small compared to the internal variability of a GCM (Sieck et al., 2016)."

L75: "RCMs" → "RCM's"

A: Will be changed accordingly.

L94: "recommend" → recommended

A: Will be changed accordingly

L97: (or L105-109) Please explain a bit more about apparent temperature. Why is it apparent? Why doesn't it always occur?

A: The term 'apparent temperature', defined by Davis et al., 2016, is a function of both temperature and dew point temperature. The dew point temperature accounts for humidity in the air. The combination of temperature and humidity is more relevant to human health than temperature alone.

L109: "Relative change" I guess you look at the change in all indices, I don't get why you specify this explicitly here.

A: Relative change in L97. The reference to 'relative change' will be removed.

L 103: "and NorESM has" → "and the NorESM driven have"

A: Will be changed accordingly.

L115-116: This sentence is incomprehensible to me. Think about what you want to say, then write that in two, or even three, sentences. Long sentences with few commas has no intrinsic value.

A: We will change the sentence to:

"A change in extreme precipitation directly influences local communities. Such communities have applied design standards for structures to withstand floods with a specified return period. These return standards will no longer be applicable when the extreme value distribution shifts with global warming."

L116: Remove "As such", this is not the correct use of that term.

A: We will delete "as such" accordingly.

L118: "exceedance probability" Isn't rainfall events rather associated with either a probability or a threshold. Maybe I just don't understand what you mean.

A: Exceedance probability for rainfall events is associated with engineering practices. In probability theory, an event can be characterised by either probability of non-exceedance, or exceedance. Here, we use the exceedance probability. We will adjust the sentence, to underline this.

L118: " rainfall intensity for the 50-year return" Do you mean rainfall intensity with a 50-year return period?

A: Yes, we mean a 'rainfall intensity with a 50-year return period'. All references to this have been changed accordingly.

L119: "Such information" What kind of information?

A: This refers to the sentence before where "rainfall intensity with a 50-year return period is computed". We will adjust the sentence to "Information on changes in the rainfall intensity with a 50-year return interval …".

L121: "between 100 and 100 years" I guess that at least on of the "100" should be another number.

A: This is a typo and should state "between 10 and 100 years"

L122: What is your definition of CDD? Is it the longest period of consecutive dry days, or something else? Is it the longest period over the whole 10-year period or is the annual average for all 10 years? What is the threshold for a dry day (1 mm?)?

A: As we explain in the paper, the threshold is less than 1 mm per day (lines 123-124). We calculated the maximum duration for the entire 10 years of each ensemble member. We will add this to the text.

L122 will be rephrased:

"Lastly, the Consecutive Dry Days (CDD), defined as the maximum number of consecutive days with a daily precipitation amount of less than 1mm over a region (Karl et al., 1999; Peterson et al., 2001) is calculated for the entire 10 year period of each ensemble member. The CDD is calculated for each of PRUDENCE regions (Christensen et al., 2007), illustrated in Figure 1, because drought indicators are relevant over large areas."

L122: Why do you analyse CDD for the Prudence regions and not in a map as with the other indices? Or, why don't you do the thorough analysis that you do for CDD for the other indices?

A: The CDD analysis is computed for the Prudence regions and not 'per-grid-box' as for the other indices used in this study, because applications drought indices are relevant over larger areas, whereas in the cases of the other indices considering high temperatures and heavy precipitation, analysis of individual grid-boxes are more relevant as they have more local applications.

L130: "historical" Is this pre-industrial (1861-1880) or current (2006-2015)?

A: This is "current" and will be changed accordingly.

L131-132: "differences of the /…/ percentiles were computed by subtracting the ensemble mean" Isn't the difference in any percentile calculated by computing the difference between the percentile for one period with the same percentile for another period? I think this sentence could be made more understandable.

A: Yes, you are right. We will change this to:

"In case of ATG28, differences of the 5th, 50th, and 95th percentiles were computed by subtracting the respective percentiles of the current decade from the projected periods."

L132: "areas" What kind of areas? Isn't it done per grid point?

A: Yes, the calculation is done on every grid point. The data points are referred to number of exceedances in the current period. The text formulation is changed to read:

"Only grid boxes with more than 20 exceedances over threshold in the current period were included in the analysis of ATG28 in order to allow for confidence interval calculations for the shown percentiles using order statistics."

L135: How is the "percentile confidence range" defined?

A: This is explained in line 134. We use order statistic to compute the confidence range for the percentiles.

L138: Why do you choose the Mann-Whitney-U-test?

A: We chose the Mann-Whitney-U-test, because it is a non-parametric test, which does not require any underlying statistical distribution.

L140: "precipitation intensity of the 50-year period" I think you mean "precipitation intensity with a 50-year return time".

A: We stated in the paper "intensity of the 50-yr return period", but in order to be more clear we will rephrase to "precipitation intensity with a 50-year return time".

L141: "historical" Is this pre-industrial (1861-1880) or current (2006-2015)?

A: This is current and will be changed accordingly.

L141: "NOResm" → "NorESM"

A: Will be changed accordingly.

L141: Why do you explicitly mention the model names here? I expect you to do analyse both ensembles for all indices. It's implicit that you do.

A: The sentence will be changed to:

"The differences in RI50yr are computed as the relative change in daily precipitation intensity of the 50-yr return period between the 1.5°C and 2.0°C simulations compared to the current period simulations."

L150: "mean temperature" Please add a "(not shown)" here.

A: This has been added accordingly:

"But also the central and eastern parts of Europe show increases in ATG28, consistent with the increase in mean temperature (not shown)."

L152: "and more in the median around the Mediterranean" Please consider rephrasing to something more understandable.

A: This has been rephrased.

"Around the Mediterranean the increase in ATG28 during the 1.5° C period is mostly moderate with up to 9 days in the median whereas changes in the 2.0°C period are reaching 18 days and more."

L154: "no change in the distribution of ATG28" Based on figs 2 & 3 I don't agree. For +2 in central Europe the 5th percentile doesn't seem to change much, while the 95th percentile increases with around 6 C. Isn't that a change in the shape of the distribution?

A: Actually, it is a shift of the distribution; we have little reason to think that the shape of the distribution is changing. This is because while we agree that the 5th percentile for ATG28 does not

change much in the simulations at 1.5 degree for the NorESM model, there are larger shifts at 2 degree, and also at 1.5 and 2 degree for the ECHAM model (Figure 3).

L154:"spatial resolution allows". It's of course better than the GCMs, but is it really true? Isn't the motivation for EUR-011 that EUR-044 doesn't resolve complex topography?

A: We agree that EUR-11 would be better to resolve complex topography, however, with the current generation of HPC computers, such a large ensemble of RCM simulations would not have been possible to conduct on much higher resolution than 0.44°.

L156"Mediterranean" → "around the Mediterranean" or "Mediterranean region"

A: This has been changed:

"This is especially important in areas with complex topography such as the Mediterranean region, which is usually only poorly resolved in GCM simulations."

Figs 2 & 3. Please consider the following: Add percentile names in a new top row. Add SWL names in a new left column. Add units by the colour bars. Add letters a-f in the caption. Add ensemble sizes in the caption And it seem like white colours are replaced by grey.

A: We will update the Figures by taking into account the comments of all referees. The grey boxes are masked out areas. On land they refer to grid boxes that do not match our criteria of 20 or more occurrences of ATG28 during the current period. This will be stated explicitly in the text and caption.

L169: "the four REMO ensemble experiments" This is a bit ambiguous. How many REMO ensembles are there, 1, 2, 4, 6? Depends on the definition. Consider erasing "four".

A: Yes, it is better to understand without "four". This will be changed to:

"Figure 4 shows the relative differences of RX5day for the REMO ensemble experiments."

L172: "more coherent" More coherent than what? Not with "larger areas".

A: The ECHAM6 driven ensemble shows a more coherent pattern than the NorESM driven ensemble. The sentence will be changed to:

"It can also be seen that the patterns in the ECHAM6 driven ensemble is more coherent than the NorESM ensemble with larger areas showing a significant change."

L172: "more significant" How is significance calculated, and how is it shown in fig 4?

A: This has been mentioned in lines 137-139. Only results at the 95% significance level are shown using a Mann-Whitney-U-test. It is missing in the caption, though. We will add this to the caption and explicitly state that grey shading refers to areas with non-significant changes.

"Figure 4. Relative difference of RX5day (in percent) between current and the 1.5°C period (left column) respectively the 2.0°C period (right column) for the NorESM with 25 members (top row) and ECHAM6 with 100 members (bottom row) driven REMO simulations. Grey shading show areas with non-significant changes on the 95% significance level."

L173: "difference in ensemble size" Between what?

A: this is referring to the difference in ensemble size between ECHAM6 and NorESM driven simulations. We will update the text.

L176:"the interior of the simulation" What is the "interior of the simulation" if not everything apart from the boundaries? This seems to be an unnecessary complicated way to describe where the largest changes are. Also consider changing "simulation" to "domain".

A: Rephrased:

"Apart from artificial effects due to the boundary conditions, the strongest signal within the core domain appears over the Baltic Sea, with an increase of up to 15% in RX5day under a 2.0°C increase in GMT."

L179: It should be easy enough to at least roughly test the effect of SST. Just plot it and see how unrealistic it is. Also check the timing of RX5day, is it in winter or in summer? I guess the SST bias works differently in different seasons. In winter it's probably too warm, in summer too cold. I suggest that you do some kind of check.

A: The formulation unrealistic is a bit misleading. It should better be formulated as "unresolved SST". If it is over- or underestimated, depends on the particular sub-basin and the surrounding (resolved) SSTs.

Figure 4. Consider the following: Add SWL names in a new top row. Add model names in a new left column. Add letters a-d to the panels. Add units to the colour bars. Explain grey shading in caption.

A: A general makeover of the plots will be made as suggested. The caption texts will be changed accordingly (see answer above).

L190: "To account for" What do you mean with this¿It doesn't seem to be the correct use of the term "to account for".

A: Sentence will be rewritten as proposed below (see response to L190ii).

L190: "the relative change in daily rainfall intensity is presented in Figure 5". No, it's not. Figure 5 shows the change in the intensity with a 50 year return time.

A: The values plotted in Figure 5 are the relative change in RI50yr given in percent. Relative change, in the case of the 1.5°C simulation, is computed as the difference in RI50yr between the 1.5°C and current period, divided by the RI50yr of the current period.

The sentence will be rephrased as:

"The relative change (in percent) in RI50yr across Europe are presented in Figure 5."

L191: "In the both the" → "In both the"

A: This will be changed accordingly.

L193: "precipitation intensity of the 50-year period" I think you mean "precipitation intensity with a 50-year return time".

A: Yes. The sentence will be reformulated to:

"ECHAM6 driven simulations clearly show increases in the 24-hour rainfall intensity with a 50-yr return time, of up to 20% over continental Europe."

L190-195: It's seems like you're struggling with how to describe the precipitation intensity of events with a 50 year return time. Why don't you just define RI50yr properly and a bit lengthy, and then just stick to RI50yr? That would save you some trouble in writing, and should avoid some confusion for the reader.

A: We agree, and will update the paragraph defining RI50yr and rewrite this paragraph using the terms accordingly.

Fig 5: Who do you suddenly show results for a different domain? Excluding parts of northern Europe and including parts of northern Africa where you don't have data. For consistency, show the same domain in all plots. This domain should preferably be the same as the model domain, unless you have a good motivation for excluding certain areas. Also, consider the following: Add SWL names in a new left column. Add letters a-d in the caption. Replace "Percent" with "%" in the legend. This is perhaps a matter of taste, but common practice is "%" I think.

A: We will update and harmonize the plots by using only one plotting tool for the horizontal plots.

L204: This sentence should start: "Both the 1.5 C and 2.0 C distributions ..."

A: We will change this accordingly

L205: "historical" Is this pre-industrial (1861-1880) or current (2006-2015)?

A: This should be "current" and will be changed accordingly.

L205: I think you can remove "respectively". It doesn't add anything.

A: We will update this sentence to: "The distributions for the 1.5°C and 2.0°C period are compared to the current CDD distribution".

L207-208: "whereas the /.../ distributions" This goes without saying. Consider removing for brevity.

A: We will delete the sentence ", whereas … distributions" here.

Table 1: Why are the p-values suddenly the most important part of the analysis of an index? And why is CDD analysed for the Prudence regions? Please explain.

A: As explained in the response to L122, we compute CDD over the Prudence regions because drought indicators are more important over large areas than on a grid-box. In addition, we are comparing distributions of CDD and therefore we need to apply a statistical test to determine whether those distributions have any significant changed. We will provide an explanation earlier within the paper as to why we analyse the CDD for the PRUDENCE regions (see answer to Line 122).

L215: "longer period of dry days" Be careful how you interpret changes in CDD. You don't define CDD so I can't be sure if you make the correct interpretation. If your CDD is averaged over your 10- year period it could be correct. If your CDD is the longest dry period over the 10-year period it only tells you that the longest dry period will be longer. That doesn't necessarily mean that dry periods on average will be longer.

A: In our response to L122, our formulation of CDD is the maximum consecutive days in 10 years. In this analysis, we are determining whether the shift in CDD distributions are significant. In Figure 6, the percentage of the entire CDD distribution of a given duration is presented.

We reformulated L215:

"Over this region, one can see an increase in duration of the longest dry period and that they occur more often (Figure 6)."

L217: ". . . indistinguishable in the simulations /.../ (Table 1)." I don't understand the this sentence. Please rewrite.

A: We will rewrite to: "In contrast, regions 6 and 8, the Alps and Eastern European region, have changes in CDD distributions that are statistically indistinguishable under 1.5 C and 2.0C compared to the current simulation (Table 1)."

L218: "more frequent" This is not correct. CDD is the length of the longest dry period (at least this is the common definition). If you want to know if dry periods will be more frequent you should study the number of dry periods.

A: By plotting the CDD distributions in Figure 6, we are studying the number of dry periods. Over region 2 one can see that the length of the longest dry period increases as the reviewer states and one can also see that the number of longer CDD increases. The authors argue 'more frequent' in this case is correct.

L221: "1.5_C vs. 2.0_C" I think yo mean "1.5_C instead of 2.0_C"

A: We agree, and will change accordingly.

L222: "adaption" Do you mean "adaptation"?

A: Will be changed accordingly

L221-222: I have a few problems with this, somewhat ambiguous, sentence. First, it is not the +1.5_C or +2_C targets that will have an impact on society, it is the climate change. Second, that the climate will be different in the +1.5_C world compared to the +2_C world is obvious. I guess you mean that the change in CDD is not linear so that the extra 0.5_C will have a large impact. Third, why do you point out changes in CDD in region 7 as a particular motivation for adaptation and mitigation? In my view this whole paper is a motivation for adaptation and mitigation as it shows how climate change may change in the future. Consider rewriting.

A: We do not state "an impact on society", we mean that there is a (positive) impact of a 1.5° target compared to a 2.0° target. Second, we show that the difference also has a measurable impact, given large natural variability for indicators such as CDD. This is not even obvious in these HAPPI simulations for several other regions, so this is in fact not straightforward.

We will rewrite the sentence to: "Thus, one can conclude for region 7, according to these simulations, a lower target of 1.5°C increase in GMT could reduce the length of the maximum number of dry days in this region, compared to a 2.0°C target."

L225-226: It's a poor motivation to exclude regions just because the U-test gives different results. Especially since the results differ also in region 3. Strictly speaking the results differ for all regions since you get different p-values (Table 1). With the possible exception of IP at +2.0 where both ensembles get 0.000.

A: We will delete this sentence, as we already provided motivation for selecting regions 2, 3 and 7 at the start of the paragraph.

Fig 6: It's very odd to measure the number of days in the unit weeks. Add to the caption something like: "for the ECHAM (top row) and NorESM (bottom row) driven ensembles".

We will add the suggested sentence to the caption.

L235: "10 x 100 years" I would prefer "100 x 10 years" since it is 100 10-year simulations.

A: This will be changed accordingly.

L250: "smaller sub-ensemble driven by ECHAM6" I don't see this sub-ensemble anywhere in the text. Should it be added to the analysis?

A: We will add this analysis as supplement material.

---

## Author Comment (AC4) · 2 Jun 2020

**Answers to Referee#2**

**General response:**

The authors would like to thank the referee for the time and effort put into this review, which has been useful in improving our manuscript. We have carefully read the comments, and provide a detailed response to all comments, below.

The paper investigates the impacts of 1.5°C and 2.0°C global warming on temperature and precipitation extremes over Europe. With this aim, the authors use an ensemble of dynamically downscaled simulations from the HAPPI project. The analysis focusses on four climate indices from ETCCDI.

The paper covers an interesting and relevant topic that could be a useful addition to the literature. Unfortunately, several aspects in terms of methods, analyses and results are unclear and need further explanation. In fact, the manuscript needs a lot of improvement in order to increase the clarity and readability. Please refer to the main points and specific comments below. In particular, the authors do not provide enough convincing evidence to prove the benefits of their method. Nevertheless, I suppose that there are sufficient arguments for it. Therefore, I suggest a major revision for this manuscript.

Answer: We will revise the introduction and methodology sections to emphasize the paper's objective of introducing a new dataset. In addition, we will increase the clarity and readability, by improving several sentences in response to the comments we received from this referee and the others. The method that the referee mentions, was developed by the HAPPI consortium. We will insert a more extensive description and motivation for this approach, thereby providing a better background. Finally, we also will provide more details on the assumptions and implications of using the HAPPI GCM simulations for regional downscaling using the regional climate model (RCM) REMO. We hope this will satisfy these main concerns of the referee.

**MAIN POINTS:**

1) Text: The whole manuscript needs a thorough proofreading and language check. Some sentences are incomprehensible; others are just too long and should be split in two to enhance readability. In addition, several phrases/descriptions are not consistent throughout the paper and therefore may confuse the reader. Please also refer to specific comments below.

A: The manuscript will be checked for language and revised to improve the flow, to include more detailed and consistent descriptions, and reduce sentence lengths. We will revise the introduction and methodology sections to emphasize the paper's objective of introducing a new dataset. Also, we use more consistent definitions for simulation periods, and other descriptions, to improve clarity.

**2) Figures: The figures and their captions need some general improvement.**

A: All Figures and captions will be improved as suggested below.

**2a) Why are some land areas in Figures 2-4 grey?**

A: The grey boxes are masked out areas. On land they refer to grid boxes that do not match our criteria of 20 or more occurrences of ATG28 during the current period. This will be stated explicitly in the text and caption in the revised paper.

2b) You should use the same format for all spatial plots, e.g. \* One colour bar including units \* Label individual plots \* Same domain \* Add more information in the figure itself (warming level, ensemble, . . .).

A: We used different plotting tools and will redo the plots with one tool following the suggestions.

2c) Figure 6: The bars are difficult to distinguish. Non-overlapping bars might be preferable. The distributions for both GCMs are very different. For NorESM, they are quite narrow, while they are much broader for ECHAM6. What are the implications of such large discrepancies? Please discuss. Why do you measure consecutive dry days in weeks?

A: This figure will be redone with non-overlapping bars. With regard to the different distribution widths, there seems to be a dependency on the forcing model. In our investigations, we see NorESM forced ensemble simulate wetter conditions, whereas ECHAM6 forced ensembles simulate warmer and drier conditions. This evaluation will be added in as supplementary material. This likely explains the differences in CDD distributions. The implication is that the absolute values coming from the models should be treated with caution, nevertheless we see a partially significant shift in CDD distributions of both ensembles. This demonstrates a qualitative (not quantitative) change towards longer dry periods. Lastly, CDD is measured in terms of weeks in the plots as it is more intuitive to interpret 8 weeks without precipitation greater that 1mm over a region than it is to read 56 days. In addition, it reduces the ink-to-data ratio in terms of visualization.

3) Is the main objective of this paper to present a new data set or to present mainly new results? Either way, both are not properly presented and need more details.

A: The aim of this paper is to present a new dataset. In addition, we provide 4 examples of how adaptation-relevant information can be derived from this dataset. We will make this clearer by improving the introduction and methodology sections. The latter will have a dedicated sub-section describing the HAPPI protocol with more details (see below).

4) Missing details/explanations: Some points (especially in the Methods section) need a better/more detailed explanation to be understandable. Your descriptions are too short and raise more questions than they answer.

A: Overall, the section on Methods will be substantially improved, as discussed above. We address below the point by point comments made on missing details and explanations also for the other sections.

4a) L58/59: Why did you use 20 years for the pre-industrial period, but only 10 years for all other simulations? From a climatological perspective, 10 years are rather short to enable a climatological view.

A: The pre-industrial period is only used as a baseline to define 0°C global mean warming. The definition is coming from the HAPPI protocol that every group doing global simulations followed. We are aware that there are several slightly different definitions of this particular period. The IPCC Special Report on 1.5°C warming lists several of these definitions. We will add a new sub-section on HAPPI to the manuscript (see below) to make these definitions more clear. Also the ten-year period is coming from the HAPPI protocol and is discussed in Mitchell et al. (2017). We will add the motivation for the ten-year period in the HAPPI sub-section.

4b) L58: Which period did you use for the future climate simulations? There are a few references given, but it should be specifically described which methods have been used here and how they are applied.

**A: The method will be explained in more detail in the sub-section about HAPPI (we included a rewritten section in the response to referee#1).**

**4c) Did you compare the simulation for a current decade to observations and/or reanalyses to check how realistic they are? This would be very important.**

A: We agree that comparing simulation results of climate models to observations is always very important to gain trust in models. The model version on this domain has been extensively evaluated using Re-analysis as boundary conditions in many other studies, and the relevant papers are already cited. In this paper, we want to demonstrate the benefits of using a large ensemble when looking at small changes in terms of global mean temperature - therefore an analysis against observations or reanalysis is less important. Especially since we only consider projected changes. We did comparisons in terms of quick views though, and concluded that the results were of comparable quality as, e.g., historical simulations from CORDEX with CMIP5 boundary conditions. However, as this point has also been raised by other referees, we will include a quick analysis on the general performance as supplement material.

**4d) L59-61: How are the warming levels (1.5 and 2.0) calculated? Did you use RCP2.6 for 1.5 warming and RCP4.5 for 2.0 C warming?**

A: The warming level has been calculated from a CMIP5 ensemble mean global mean temperature response. In case of the 1.5°C period RCP2.6 was used. The 2.0°C period is calculated using a weighted mean between RCP2.6 and RCP4.5. A more extensive explanation is given in the new subsection on the HAPPI experiment (please refer to response to referee#1).

**4e) L73: The regional ensemble consists of 125 members, correct?**

A: We have 125 members (100 from ECHAM6 and 25 from NorESM) for each of the three periods (current, 1.5° and 2.0° periods).

**4f) L122-124: How exactly did you define CDD? Why did you calculate the CDD for the PRUDENCE regions and not on grid-point basis as the other indices? Is CDD the maximum number per year or over the 10-year period?**

A: As we explain in the paper, the threshold is less than 1 mm per day (lines 123-124). We calculated the maximum duration for the entire 10 years of each ensemble member. The CDD analysis is computed for the Prudence regions and not 'per-grid-box' as for the other indices used in this study because applications drought indices are relevant over larger areas, whereas in the cases of the other indices considering high temperatures and heavy precipitation, analysis of individual grid-boxes are more relevant as they have more local applications. We will add this to the text.

L122 will be rephrased: "Lastly, the Consecutive Dry Days (CDD), defined as the maximum number of consecutive days with a daily precipitation amount of less than 1mm over a region (Karl et al., 1999; Peterson et al., 2001) is calculated for the entire 10 year period of each ensemble member. The CDD is calculated for each of PRUDENCE regions (Christensen et al., 2007), illustrated in Figure 1, because drought indicators are relevant over large areas.

**4g) L130-145: Why did you choose different statistical methods to investigate the individual climate indices?**

A: It is a common procedure to employ different statistical methods for different climate indices depending on the physical variables from which it computed from, for example, temperature or precipitation. This is because some statistical tests assume a given underlying distribution shape, for example temperature generally follows a normal distribution, whereas precipitation does not. Thus,

different statistical tests ought to be employed for climate indices derived from temperature versus precipitation. In addition, the use and application of the different indices also warrant different approaches and methods.

4h) L130-145: The application of the significance measures is unclear, please rewrite. There are two methods used for two different parameters (ATG28 and RX5day). There seems to be some confusion on what is used for CDD (L143 says CDD similar to ATG28, but the method for CDD seems to be similar to RX5day, namely Mann-Whitney). No information is given for RI50yr. Anyway, the paper provides only information on the significant changes for CDD, but not for the other parameters. This should be remedied.

A: We will rewrite section 2.3 and dedicate one sub-section to each indicator to better distinguish the used methods between them.

4i) Why do you think that all differences between the two ensembles are due to the different ensemble sizes (e.g. L172-175)? They could also result from the driving GCMs. It might be useful to include the results for the smaller sub-sample of ECHAM6 that you mentioned in the text.

A: As we stated already in the paper, analysed this with a random sub-set of 25 ECHAM6 members and found similar, noisier patterns like in the NorESM ensemble. This supports our conclusion regarding ensemble size. However, to provide this actual information to the readers, we will add the ECHAM6 sub-set plots as supplement, in the revised paper.

4j) Table 1: Partly, the smaller ensemble (NorESM) generates more significant results. How does this relate to the hypothesis that a larger ensemble size is beneficial?

A: We disagree with the reviewer, as it cannot be concluded on the basis of the CDD in Table 1 only that the NorESM smaller ensemble provides more robust results. This could be by chance, as the NorESM model could lead to relatively dry projections, for instance, leading to changes that are more significant. We have looked at four different indices (precipitation and temperature), and across the board the larger ensemble has less noise, regardless of the change, at the same level of warming.

4k) Compare your results to previous studies (see interactive comment by Laura Suarez-Gutierrez for more details).

A: We have provided responses to the comments from Laura Suarez-Gutierrez. We will include comparisons to other studies, but many of Laura Suarez-Gutierrez suggestions are beyond the scope of our paper. Please see our responses to her comments.

4I) The authors should not oversell their results (or should argue more convincingly). E.g. Impact of the ATG28 increase (L241-246): What does such a change really mean w.r.t health issues? I assume that the number of days above the threshold is already high around the Mediterranean? Does a change of O(10days) drastically change the base level and/or the potential health impacts in this region? Furthermore, is a resolution of 50km sufficient to derive change estimates for local adaptation measures? The authors do not provide enough convincing evidence for the benefits of their method. Nevertheless, I suppose that there are sufficient arguments for it, but this should be better phrased.

A: The chosen threshold is relevant in particular for sudden cardiac death exposure. The number of days in the baseline is of similar order. Please bare in mind that this indicator is not dependant on temperature alone, but that humidity also plays an important role. We agree that 50km might not be sufficient to inform local adaptation measures, especially if one thinks about cities. But in this regard

our results would serve as a lower bound of expected changes, because we have no urban heat island effects in our model. We will rework the discussion section to make it more convincing.

**SPECIFIC COMMENTS**

**Data set, dataset or data-set?**

A: We will use the term dataset and change the manuscript accordingly.

**NOResm or NorESM?**

A: We will use the term NorESM and change the manuscript accordingly.

**L17: "measures at a" - Delete "a".**

A: We will correct as recommended.

**L20-22: This sentence is incomprehensible, especially the first part.**

A: We will rephrase that sentence as follows:

"Identifying regional climate change impacts for different global mean temperature targets is increasingly relevant to both the private and public sectors. In the private sector, investors demand financial disclosure associated with climate change risks and opportunities (Goldstein, et al., 2018). In the public sector, policy makers rely on climate information build on internationally agreed limits to develop national climate action policies."

**L25: "current generation global climate simulations" -> "current generation of global climate simulations"**

A: We will change as recommended.

**L45-47: Please rephrase sentence.**

A: We will merge the sentence with the following sentence:

"Here, we develop two regional climate datasets of 25 and 100 members to create a large ensemble of RCM simulation which are particularly suitable to study extremes. Earlier studies such as Leduc et al. (2019) have successfully demonstrated the usefulness of such an approach."

**L55-56: You cannot downscale an RCM using GCM simulations.**

A: We will rephrase that sentence:

"To create a data set for regional climate impact studies for Europe under 1.5°C and 2.0°C global warming the regional climate model REMO has been used to dynamically downscale two GCM ensembles following the HAPPI experiment protocol by Mitchell et al. (2017)."

**L59: What does "CMIP5 mean SST anomaly" actually mean? All CMIP5 models and members, or just some, only ECHAM6/NorESM? Is there a reference?**

A: This point will be addressed in a dedicated section explaining the HAPPI protocol in more detail (see answer to referee#1).

In the HAPPI protocol all CMIP5 models are included in the averaged SST for the current and the projected periods of 1.5°C and 2.0°C respectively. The SST anomalies of the 1.5°C projected period are calculated by subtracting the averaged current SST from the averaged SSTs of the 1.5°C

**projection. The SST anomalies are then added to the observed SSTs of 2006-2015. The 2.0°C SST anomaly is computed in a similar manner.**

**L66/67: "from the core domain defined by CORDEX the entire domain has 121x129 grid boxes" – Please reword.**

A: We will rephrase:

"The European CORDEX domain for REMO covers 121x129 grid boxes. To exclude the sponge zone, where the REMO simulations are relaxed towards the GCM solutions, a core domain of 106x103 grid boxes, following the CORDEX definition, is used for the analyses."

**L94: "recommend" -> "recommended"**

A: We will changed as recommended.

L97-100: You defined an abbreviation for each climate index. Use them more consequently throughout the text.

A: We will make changes made as recommended.

L99: "precipitation intensity at the 50-yr return period" – Do you mean precipitation intensity with a 50-yr return period? Please use a consistent explanation throughout the text.

A: Yes, we mean precipitation intensity that occurs every 50 years. We will provide an explanation, and use the definition more consistently throughout the text.

**L111: What do you mean with "annual sum maximum"?**

A: This is a spelling mistake. It should be "annual sum". We will revise:

"The index for the annual maximum of the five-day precipitation sum (RX5day) is used to characterise heavy precipitation events, which can be relevant for flood generation in river basins.

**L121: "between 100 and 100 years" – Please correct.**

A: This will be corrected to '10 and 100 years'.

L130: What do you mean with "historical"? Pre-industrial or current? Same in L141, L205, L215.

A: This is an inconsistency. It should be "current" everywhere. The text will be changed accordingly.

**L132: "Only areas with more than 20 non-zero data points" – Isn't the calculation done at every single grid point?**

A: Yes, the calculation is done on every grid point. The data points refer to number of exceedance in the current period. The formulation will be changed:

"Only grid boxes with more than 20 exceedances over threshold in the current period were included in the analysis of ATG28 in order to allow for confidence interval calculations for the shown percentiles using order statistics."

**L152: "in the median around the Mediterranean" – Please reword.**

A: This will be rephrased as follows:

"Around the Mediterranean the increase in ATG28 during the 1.5° C period is mostly moderate with up to 9 days in the median whereas changes in the 2.0°C period are reaching 18 days and more."

**L154: Is a spatial resolution of 0.44 high enough to resolve complex topography? What about EUR-11?**

A: We agree that EUR-11 would be better to resolve complex topography, however, with the current generation of HPC computers, such a large ensemble of RCM simulations would not have been possible to conduct on much higher resolution than 0.44°.

**L176: "interior of the simulation domain" – Please reword.**

A: We will rephrase:

"Apart from artificial effects due to the boundary conditions, the strongest signal within the core domain appears over the Baltic Sea, with an increase of up to 15% in RX5day under a 2.0°C increase in GMT."

**L190-191: Please reword sentence.**

A: We will rephrase as follows:

"The relative change (in percent) in RI50yr across Europe are presented in Figure 5."

**L204: You never defined/explained p-values.**

A: Please see our answer to L204 below

**L204: "Both the distributions 1.5C and 2.0C" – Please reword.**

A: We will rephrase the entire section to read as follows:

"In this section, the changes in the Consecutive Dry Days (CDD) distributions for the 1.5°C and 2.0°C periods compared to the current period are presented. To distinguish whether these changes in the distributions are statically significant we employ the Mann-Whitney U-Test. Where the resulting p-values of the test are greater or equal to a significance level, alpha, of 0.05 or smaller, the null hypothesis is rejected indicating the distributions differ. The p-values for each of the PRUDENCE regions (Christensen et al., 2007), shown in Fig. 1, are presented in Table 1."

L215: "shift towards longer periods of dry days" – This depends on your definition of CDD. If CDD is the maximum number of consecutive dry days, your results only show that the longest dry period is getting longer in a warmer climate. That does not mean that all dry periods will be longer.

A: Our formulation of CDD is the maximum consecutive days in 10 years. In this analysis, we are determining whether the shift in CDD distributions are significant. In Figure 6, the percentage of the entire CDD distribution of a given duration is presented.

**L215 reformulation:**

"Over this region, one can see an increase in duration of the longest dry period and that they occur more often (Figure 6)."

L216: "the Alps and Eastern European region" -> "the Alps and Eastern Europe"

**A: We will rephrase accordingly.**

L218: "suffer from more frequent and longer drought periods" – Again, this depends on your definition of CDD. If you used the common one (CDD being the length of the longest dry period), you cannot say anything about the frequency of dry periods.

A: In Figure 6, we have plotted the distribution of CDD and are thereby studying the frequency of dry periods. Over region 2 one can see that the length of the longest dry period increases as the reviewer states and one can also see that the number of longer CDD increases. The authors argue 'more frequent and longer drought periods' in this case is correct.

**L222: "adaption" should be "adaptation" Figure 6: Use ECHAM6 instead of ECHAM.**

A: We will change accordingly.

L235: "10 x 100 years" -> "100 x 10 years"

A: We will hange accordingly.

**L247: "RX5day" -> "(RX5day)"**

A: We will change the sentence to:

"The RX5day shows a general increase over Europe which is more pronounced under higher global mean warming."

L251: "such precipitation extremes" -> "such as precipitation extremes"

A: "as" will be included.

L263: "historically similar" – Strange wording.

A: See our answer to L263-264, below.

L263-264: "pre-industrial period" – I thought you were calculating differences between future and current climate and not between future and pre-industrial?

A: There was a mix-up in the sentence. It should always be current vs. future. The text will be changed accordingly, as:

"The changes to CDD distributions show that Spain will experience significantly more drought conditions in the future compared to the current period, even at a 1.5°C increase in GMT. For Italy, drought conditions associated with the 1.5°C simulations show non-significant changes, yet those associated with the 2.0°C simulations are significantly different to the current period, thus showing possible consequences of exceeding the 1.5°C GMT target of the Paris agreement."

---

## Author Comment (AC5) · 2 Jun 2020

**Answers to Referee#3**

General response:

The authors would like to thank the referee for the time and effort put into this review. These comments have been useful in improving our manuscript. We have carefully read the comments and provide a detailed response to each comment.

This paper presents results from a regional dynamical downscaling of two GCM ensemble from the HAPPI project and compares the model output to 4 extreme event indices.

The large number of ensemble members and important can potentially make a good contribution in how to analyse and estimate the difference between relatively close climate change "targets".

One major discrepancy however is that there are no evaluation of the quality of results with respect to observations or reanalysis. If the model results deviates to much from the "real world" one can not trust the conclusions given for the warm periods. The discussion does not have to be very advanced but I think you should include a point on this.

Answer: We did a quick analysis on this and found that the results are in good agreement with other downscaling activities using REMO for Europe (e.g., CMIP5 downscaling of GCMs). As suggested to referee#2, we will add a short analysis as supplement.

In general both the description of methods and results are not always very clearly defined. Sometimes it is just a matter of language. The sentences are sometimes so long that the reader loose the thread before getting to the end, so please try to be more focused.

A: We re-formulated many sentences that caused confusion in response to Reviewer#1 and #2.

A problem is that it is sometimes unclear whether the assumptions used are your / RCM limitations or inherited from the Happi protocol. Event hough I know the Happi protocol it was sometimes hard to separate. I see that many of the questions from other reviewers on critical assumptions e.g. how does pre-industrial come into play, when is the future time slice .. is often related to the protocol. I think the authors should include a summary of this information, in particular as you rightly point out that the Happi protocol is quite different from the traditional CMIP settings.

A: We came to the same conclusion. Therefore, we separated the description of the HAPPI protocol from the REMO model set-up and extended it (see answer to Reviewer#1).

The defintions/ presentations of the indices and how they are used can be improved (Section 2.2 and 2.3) It is particularly hard to follow the setup of significance tests. "RX5 days are computed ...similar to ATG28, however …

A: We will adjust the text and add sub-sections for each indicator.

CDD is similar to ATG28 but with the method used for RX5 but now however? There are no significance test for RI50yr ? There does not have to be one but I would like to know.

A: No, we did not use a significance test for RI50yr.

With only 4 indices I think you should avoid "similar" as much as possible and just describe the methods for each.

A: Separating the description of the indicators should make it more clear (see answer above).

The discussion on being able to reproduce the more noisy results of the smaller ensemble by picking a smaller number of the largest ensemble was interesting. I wonder if it is any way of presenting this in a cumulative manner, with respect to number of ensemble members. The change rate may also indicate whether even the largest ensemble is too small. I realize however that this is likely beyond the scope of the article.

A: We will include an analysis on the randomly picked 25 member ensemble from ECHAM6 as a supplement as suggested to Reviewer#2. We agree the further analysis on the change rate would be beyond the scope of this article.

To the point on data availability: Is it the data used specifically for this study, or is it general RCM output.

A: We are still struggling with finding a data centre that can host the data. Due to CMIP6 activities our dataset is of low priority. However, we managed to cmorize the most common variables and can provide them by request. This is already ongoing in the framework of other projects we are working on.

Some minor suggestions /corrections

Line 20: "differ" not needed

A: We will rephrase this sentence to (see also answer to Reviewer#1):

"Identifying regional climate change impacts for different global mean temperature targets is increasingly relevant to both the private and public sectors."

Line 25: Include a line on scenario definition in CMIP6?

A: We will add CMIP6 to this sentence:

"Temperature targets, however, are not directly related to the Representative Concentration Pathways (RCP, Van Vuuren et al., 2011) used in the experimental design of CMIP5 (Taylor et al., 2012) or the Shared Socioeconomic Pathways (SSP, Meinshausen et al., 2019) used in CMIP6 (Eyring et al., 2016)."

Eyring, V., Bony, S., Meehl, G. A., Senior, C. A., Stevens, B., Stouffer, R. J., and Taylor, K. E.: Overview of the Coupled Model Intercomparison Project Phase 6 (CMIP6) experimental design and organization, Geosci. Model Dev., 9, 1937–1958, https://doi.org/10.5194/gmd-9-1937-2016, 2016.

Meinshausen, M., Nicholls, Z., Lewis, J., Gidden, M. J., Vogel, E., Freund, M., Beyerle, U., Gessner, C., Nauels, A., Bauer, N., Canadell, J. G., Daniel, J. S., John, A., Krummel, P., Luderer, G., Meinshausen, N., Montzka, S. A., Rayner, P., Reimann, S., Smith, S. J., van den Berg, M., Velders, G. J. M., Vollmer, M., and Wang, H. J.: The SSP greenhouse gas concentrations and their extensions to 2500, Geosci. Model Dev. Discuss., https://doi.org/10.5194/gmd-2019-222, in review, 2019.

Line 33:skip the word "indeed"

A: We will change the sentence to:

"The high natural variability in models requires the creation of large ensemble datasets (Deser et al., 2013)"

line 70: Green-house-gas -→greenhouse gas

A: This will be changed accordingly.

line 81: Sea-Ice → sea-ice (usually not capital letters)

A: This will be changed accordingly.

line 121: Missing intervall ("100" years on both sides)

A: This is a typo and should read "between 10 and 100 years".

Figure 1. Is the outer map equal to the model domain?

A: Yes, the outer domain is equal to the model domain.

Line 135-136 The explanation is fine, but then you do not need to mask it out either. It would show up as insignificant?

A: We agree that masking "non-significance" and masking for other reasons is confusing. We will change that in an updated version of the Figure. We masked the SST before running the test. Hence, we do not know if it would show significance or not.

Line 141 NOResm → NorESM

A: Will be changed accordingly.

Figure 6. The difference between the two downscaling sets are quite large. Any comments.

A: This is related to the forcing model. On the one hand, the NorESM ensemble is wetter compared to the ECHAM6 ensemble. On the other hand, the ECHAM6 driven ensemble seems to be biased towards warm/dry conditions. This can be examined in more detail when a small evaluation is added to the supplement.

line 260 -265 use significance instead of unlike / similar

A: We will change the text to the following formulation (see also response to referee#2):

"The changes to CDD distributions show that Spain will experience significantly more drought conditions in the future compared to the current period, even at a 1.5°C increase in GMT. For Italy, drought conditions associated with the 1.5°C simulations show non-significant changes, yet those associated with the 2.0°C simulations are significantly different to the current period, thus showing possible consequences of exceeding the 1.5°C GMT target of the Paris agreement."

---

## Referee Report (RR1)

I would like to thank the authors for considering the comments from me and the other reviewers. The inclusion of an description of the HAPPI protocol explains a lot and will avoid future misunderstandings. Unfortunately, the new text introduces some new unclear points that have to be resolved; and the paper doesn't make the impression of being carefully read through.

Mainly, there are two unresolved things: the shift in distribution of ATG28 and the frequency of dry periods. I, and other reviewers, commented on this in the last review, but I don't think the authors could motivate well enough in their response why they didn't agree with our comments. If the authors don't want to change the text they should be able to motivate why, and explain how the analysis supports their conclusions.

I don't want to come across a unnecessary disagreeable, but as a reviewer I am, in a way, responsible for this paper and that it is of good quality when its published. Comments follow below:

L81-82: "a weighted sum of RCP2.6 and RCP4.5 is calculated with a global mean temperature response of 2.05°C"
I don't understand this. What is weighted? Two time slices of RCP2.6 and RCP4.5? but RCP2.6 never reaches 2.05. Is it the sum of RCP2.6 and RCP4.5 at the end of the century? What is the level of warming in RCP4.5 at the end of the century? How is the sum weighted?

L83: "sea ice extend" → "sea ice extent"

L127-128: "All four climate indices are calculated from the daily mean precipitation, temperature, and/or dew-point temperature output of the model; for each year and ensemble member"
All four indices are **not** calculated from the daily mean precipitation, temperature, and/or dew-point temperature. I would suggest changing to: "All four climate indices are calculated for each year and ensemble member"

L164: "For each of the climate indices /.../ computed as follows". Since the change of the indices are calculated in different ways, I don't think this is the best way to start this section. I would go for something like: "The future change of the climate indices are computed as follows:"

L165-166: This seems to be a very complicated way to explain what you are doing. Why not just? "For ATG28 we calculate the differences of the 5$^{th}$, 50$^{th}$ and 95$^{th}$ percentiles between the current period and the projected periods".

L166-167: "20 exceedances" Per month, year, or in the whole 10 years?

L172: RX5day also show relative change (Fig. 4), it is not computed as a simple subtraction, and not similar to ATG28. Don't use "similar to ATG28" it is only confusing.

L177-178: Mann-Whitney is not mentioned in the description of ATG28, but it is in RX5day. Either mention Mann-Whitney in the description of ATG28, if you use it, or change to RX5day. Don't use "similar to ATG28" it is only confusing.

L184: Also add an explanation of grey boxes over ocean.

L189: "no shift in the distribution". Maybe I just don't understand what you mean, but I still don't see this. When I look at figs 2 d-f and 3 d-f I see that in some areas the 95$^{th}$ percentile increases more than the 50$^{th}$ and 5$^{th}$ percentiles. To me this means, not only a shift in the distribution, but also a change in the shape of the distribution. I made a comment about this in my previous review. That you "have little reason to think that the shape of the distribution is changing" is not a satisfactory reply to that. I see a reason that the shape is changing: for temperature extremes warming is not linear and it is known that different parts of the distribution changes in different ways. A recent

example of this, using HAPPI data, is Lewis et al. (2019) (https://www.sciencedirect.com/science/article/pii/S2212094719300556). They identify temperature hotspots where the tail of the temperature distribution increases with mean land surface warming at a faster rate than the rest of the temperature distribution. Two of these hotspots are central Europe and the Mediteranean.

It would be easy for you to calculate the distance between the 5$^{th}$ and 95$^{th}$ percentiles for the current, 1.5 and 2 simulations respectively. That would give you some indication.

Fig 2 Explain grey areas over land. Change to a figure of higher resolution in the next version.

Fig 3: See comments on Fig 2.

Fig 4: Change to a figure of higher resolution in the next version.

L225: What do mean by "To account for"?

L238: "statically" → "statistically"

L239-240: This is a highly confusing sentence. It seems like you are comparing your p-value to 0.05 or a number smaller than that (alpha), and if the p-value is greater than that the difference is significant. However, if alpha is much smaller than 0.05, for example 0, all p-values will be greater. Also, you should have a fixed alpha, either its equal to 0.05 or something else. Furthermore, shouldn't your p-value be **smaller** than 0.05 to be significant? Consider rewriting to something like: "When the resulting p-values of the test are smaller than or equal to a significance level of 0.05, the null hypothesis is rejected indicating that the distributions differ."

L242-243: This is repeating what is written above, remove.

L250: "1.5°C vs. 2.0°C" Do you mean "1.5°C and 2.0°C"?

L252-254: I still don't agree that you can conclude that the dry periods will occur more often, but after going through the reviewers comments and the responses again I start to see where the misunderstanding comes from. Remember that you only have the longest dry period for each grid point, you don't know how many dry periods you have. Therefore you can't say anything about the frequency of the dry periods. It could be that a prolongation of the longest dry period means that all dry periods will be longer (same or increased frequency). It could also be that several short dry period are replace by one long (decreased frequency). We don't know that. You could, however, argue that within a region longer dry periods will be more probable. I suspect this is what you mean, correct me if I'm wrong.
Let's take the example of the Iberian Peninsula in ECHAM. It's clear that there is a shift in the distribution between "hist" and "2.0". The longest dry period will be longer, and the chance that a gridpoint will experience a dry period of, let's say, 10 weeks is much bigger in "2.0" than in "hist". Does this mean that longer dry periods will occur more often (more frequent)? Not necessarily. You could say that its more likely in the future that the longest dry period somewhere on the Iberian Peninsula will be longer than 10 weeks, than today. The common definition of frequency is how often something happens in time, not in space. Remember also that the different gridpoints are not independent. It's likely that several of the longest dry periods in different gridpoints occur at the same time. It could be that **all** of the longest dry periods occur at the same time.

L253: I think correct English is "indistinguishable from" not "indistinguishable compared to"

L253-254: This is a ambiguous sentence. It's true that you can deduce that region 2, will suffer from more frequent and longer drought periods than experienced before.  But compared to regions 6 and 8? Even if the increase in 2 is larger than in 6 and 8, the dry periods in 6 and 8 could still be longer than in 2 (I know it isn't, but still). I guess what you're after is that the longest dry period changes more in some regions (e.g region 2) and less in others (e.g. regions 6 and 8).

L256-258: With the same kind of argument you could say that the British Isles would benefit from an temperature increase of 2 instead of 1.5. From Fig 6 its obvious that several regions would benefit from a lower temperature increase. Wouldn't you agree that the Iberian Peninsula would benefit more than the Mediterranean from a reduce warming? Although the change at 1.5 is already significant.

L287: Please insert "RI50" somewhere here.

---

## Referee Report (RR2)

Review of Weather extremes over Europe under 1.5 °C and 2.0 °C global warming from HAPPI regional climate ensemble simulations

The paper has already gone through one round of review and in particular the description of methodology is in much better shape now than the initial submission.

The duality of the paper trying to both presenting the data set and examples on how it can be used still detracts somewhat from both.
Although I realize that this is likely beyond the scope of the paper I think the analysis would have been more interesting if the indices were also compared to observations and/or reanalysis.

The presentation of the data set itself can however be easily improved either through writing that you follow a certain standard, e.g. CORDEX and / or write a brief description in the supplement. What are the main variables and output frequency.
Are they available for others? The availability notice says where they are stored, but not if they can be accessed.

Since the resolution of the RCM is still quite coarse and not that different from the GCM (0.4 vs 1 degrees) I also think it would be useful if the authors discuss if they expect the conclusions to stay the same for higher RCM resolution. Although the pattern will likely be more noisy would e,g, 25 simulations with 0.22 degree resolution look like the 25 member subset?
In particular many of the important meso-scale precipitation systems in the Mediterranean region are not resolved in a model with 0.4 degrees resolution.

Specific comments:

line 3. "Dataset" is used two times in the same line
line 70. anomaly → increment?

Line 90-95: Add information on GCM resolution.

Line 97. How are the land use described in REMO for future scenarios?'

Line 114. "one initial soil temperature state for every ensemble member in one period." But moisture profile is can vary?

Line 139 "The index for the annual maximum. → minor quibble. Do you call it an index or can you just write  The annual maximum …

Figure 1: Perhaps obvious but can not see if  IP is a subsection of MD (I presume it is)

line 215 "because the GCMs usually do not resolve these small basins. In these locations the SST is interpolated from the nearest SST value of the GCM, which might not be adequate for the region"

The basin is missing entirely? A larger grid size should just reduce the ocean fraction of the grid?

---

## Referee Report (RR3)

Comments ESD 2020-4 v. 2

I thank the authors for taking all reviewer comments seriously and changing the text accordingly. The manuscript is in much better shape now. Track changes has left some missing spaces, double spaces, double periods etc. Check carefully again. I also find some grammar mistakes, I wont go through them here, please check carefully. I trust ESD to make some additional language check.

There are still some small thing that I would like the authors to consider before publishing. There is no line numbering in the new version, I'll try my best to reference parts of the properly.

Mitchell formula:

Do "RCP2.6" and "RCP4.5" represent temperature here? Maybe it would be wise do denote them something like $T_{RCP4.5}$ or T(RCP4.5) to avoid confusion with the emissions or radiative forcing usually associated to RCPs.

Maybe it's just me, but I don't understand how 2.05 is exactly 0.5 more than 1.5.

Figure 1: Maybe explain what a sponge zone is.

Figure 2: Still a bit blurry

Figure 3: Still a bit blurry

"this can lead to biases in heavy precipitation amounts.." Remove last dot.

Figure 4: Still a bit blurry

Figure 5: Still a bit blurry

"indicating that the distributions differ. The p-values for each of the PRUDENCE regions (Christensen and Christensen, 2007), shown in Fig. 1, are presented in Table 1. Bold numbers in Table 1 indicate that the distributions differ according to the test."

Could be changed to

"indicating that the distributions are significantly different. The p-values for each of the PRUDENCE regions (Christensen and Christensen, 2007, shown in Fig. 1) are presented in Table 1. Bold numbers in Table 1 indicate that the distributions are significantly different."

"one can see an increase in duration of the longest dry period" ->

"one can see an increase in the duration of the longest dry period"

"The ECHAM6 simulations suggest there is no difference" ->

"The ECHAM6 simulations suggest that there is no difference"

Figure 6: Still a bit blurry

---

## Author Response (AR2)

**Reviewer 1**

I would like to thank the authors for considering the comments from me and the other reviewers. The inclusion of an description of the HAPPI protocol explains a lot and will avoid future misunderstandings. Unfortunately, the new text introduces some new unclear points that have to be resolved; and the paper doesn't make the impression of being carefully read through. Mainly, there are two unresolved things: the shift in distribution of ATG28 and the frequency of dry periods. I, and other reviewers, commented on this in the last review, but I don't think the authors could motivate well enough in their response why they didn't agree with our comments. If the authors don't want to change the text they should be able to motivate why, and explain how the analysis supports their conclusions.

I don't want to come across a unnecessary disagreeable, but as a reviewer I am, in a way, responsible for this paper and that it is of good quality when its published. Comments follow below:

A: We thank the reviewer for the additional comments, and we appreciate the opportunity to clarify several issues and improve the paper.

L81-82: "a weighted sum of RCP2.6 and RCP4.5 is calculated with a global mean temperature response of 2.05°C" I don't understand this. What is weighted? Two time slices of RCP2.6 and RCP4.5? but RCP2.6 never reaches 2.05. Is it the sum of RCP2.6 and RCP4.5 at the end of the century? What is the level of warming in RCP4.5 at the end of the century? How is the sum weighted?

A: It is a weighted sum of the multi-model global mean temperature responses between RCP2.6 and RCP4.5. Mitchell et al. (2017) used the following formula W1 × RCP2.6 + W2 × RCP4.5 with W1 = 0.41 and W2 = 0.59, which results in a 2.05°C global mean temperature response. We have changed the sentence to:

"Therefore, Mitchell et al. (2017) calculated a weighted sum of the RCP2.6 and RCP4.5 multi-model is calculated with a global mean temperature response using the following formula: W1 × RCP2.6 + W2 × RCP4.5 with W1 = 0.41 and W2 = 0.59. The results adds up to of 2.05°C, which is exactly 0.5°C more compared to the chosen 1.5°C period."

L83: "sea ice extend" → "sea ice extent"

A: Changed accordingly.

L127-128: "All four climate indices are calculated from the daily mean precipitation, temperature, and/or dew-point temperature output of the model; for each year and ensemble member" All four indices are not calculated from the daily mean precipitation, temperature, and/or dew-point temperature. I would suggest changing to: "All four climate indices are calculated for each year and ensemble member"

A: Changed as suggested

L164: "For each of the climate indices /.../ computed as follows". Since the change of the indices are calculated in different ways, I don't think this is the best way to start this section. I would go for something like: "The future change of the climate indices are computed as follows:"

A: Changed as suggested.

L165-166: This seems to be a very complicated way to explain what you are doing. Why not just? "For ATG28 we calculate the differences of the 5th, 50th and 95th percentiles between the current period and the projected periods".

A: Changed as suggested.

L166-167: "20 exceedances" Per month, year, or in the whole 10 years?

A: This refers to 20 exceedances during the current period, which corresponds to the entire 10 years.

L172: RX5day also show relative change (Fig. 4), it is not computed as a simple subtraction, and not similar to ATG28. Don't use "similar to ATG28" it is only confusing.

A: We deleted this part and split the sentence into two:

"Differences for RX5day are computed by subtracting the ensemble mean of the current decade from the 1.5°C and 2.0°C periods. Statistical significance for RX5day was calculated using a Mann-Whitney-U-test and only results are shown with a significance at the 95% level."

L177-178: Mann-Whitney is not mentioned in the description of ATG28, but it is in RX5day. Either mention Mann-Whitney in the description of ATG28, if you use it, or change to RX5day. Don't use "similar to ATG28" it is only confusing.

A: The sentence is changed to:

"Differences in the distribution of CDD are calculated with a Mann-Whitney U-Test with a significance at the 95% level, determining whether samples from the two periods are drawn from a population with the same distribution."

L184: Also add an explanation of grey boxes over ocean.

A: Added one sentence:

"Ocean boxes are masked out, because any change is very closely related to the prescribed SSTs."

L189: "no shift in the distribution". Maybe I just don't understand what you mean, but I still don't see this. When I look at figs 2 d-f and 3 d-f I see that in some areas the 95th percentile increases more than the 50th and 5th percentiles. To me this means, not only a shift in the distribution, but also a change in the shape of the distribution. I made a comment about this in my previous review. That you "have little reason to think that the shape of the distribution is changing" is not a satisfactory reply to that. I see a reason that the shape is changing: for temperature extremes warming is not linear and it is known that different parts of the distribution changes in different ways. A recent example of this, using HAPPI data, is Lewis et al. (2019) (https://www.sciencedirect.com/science/article/pii/S2212094719300556). They identify temperature hotspots where the tail of the temperature distribution increases with mean land surface warming at a faster rate than the rest of the temperature distribution. Two of these hotspots are central Europe and the Mediteranean. It would be easy for you to calculate the distance between the 5th and 95th percentiles for the current, 1.5 and 2 simulations respectively. That would give you some indication.

A: We think you are referring to the line: "there is no change in the shape of the distribution" and thank you for pointing this out. A (significant) shift can be seen everywhere, where there are yellow to red colours. We computed the distance between the percentiles as you requested. For many areas, we only see changes of +-1 or 2 days. But indeed we can see a change in shape for the

Northern part of Spain, the French Atlantic Coast and parts of Eastern Europe. Will change the sentence to:

"For the Northern parts of Spain, at the French Atlantic coast and parts of Eastern Europe, we can note stronger changes for the 95th compared to the 5th percentile. This means that the shape of the distribution changes towards more high extreme values. For most of the other regions there is only little or no change in the shape of the distribution of ATG28."

Fig 2 Explain grey areas over land. Change to a figure of higher resolution in the next version.

A: Explanation added. Better resolved versions are available. We apologize for the poor quality in the draft version.

Fig 3: See comments on Fig 2.

A: See above

Fig 4: Change to a figure of higher resolution in the next version.

A: See above

L225: What do mean by "To account for"?

A: We rephrased the sentence to:

"Figure 5 shows spatial differences the in 50-year return period of daily rainfall intensity (RI50yr) across Europe"

L238: "statically" → "statistically"

A: Changed accordingly

L239-240: This is a highly confusing sentence. It seems like you are comparing your p-value to 0.05 or a number smaller than that (alpha), and if the p-value is greater than that the difference is significant. However, if alpha is much smaller than 0.05, for example 0, all p-values will be greater. Also, you should have a fixed alpha, either its equal to 0.05 or something else. Furthermore, shouldn't your p-value be smaller than 0.05 to be significant? Consider rewriting to something like: "When the resulting p-values of the test are smaller than or equal to a significance level of 0.05, the null hypothesis is rejected indicating that the distributions differ."

A: We have simplified the sentence as suggested.

L242-243: This is repeating what is written above, remove.

A: We removed the sentence, which was accidentally repeated.

L250: "1.5°C vs. 2.0°C" Do you mean "1.5°C and 2.0°C"?

A: Yes. Changed as suggested.

L252-254: I still don't agree that you can conclude that the dry periods will occur more often, but after going through the reviewers comments and the responses again I start to see where the misunderstanding comes from. Remember that you only have the longest dry period for each grid point, you don't know how many dry periods you have. Therefore you can't say anything about the frequency of the dry periods. It could be that a prolongation of the longest dry period means that all dry periods will be longer (same or increased frequency). It could also be that several short dry period are replace by one long (decreased frequency). We don't know that. You could, however,

argue that within a region longer dry periods will be more probable. I suspect this is what you mean, correct me if I'm wrong. Let's take the example of the Iberian Peninsula in ECHAM. It's clear that there is a shift in the distribution between "hist" and "2.0". The longest dry period will be longer, and the chance that a gridpoint will experience a dry period of, let's say, 10 weeks is much bigger in "2.0" than in "hist". Does this mean that longer dry periods will occur more often (more frequent)? Not necessarily. Youcould say that its more likely in the future that the longest dry period somewhere on the Iberian Peninsula will be longer than 10 weeks, than today. The common definition of frequency is how often something happens in time, not in space. Remember also that the different gridpoints are not independent. It's likely that several of the longest dry periods in different gridpoints occur at the same time. It could be that all of the longest dry periods occur at the same time.

A: We would like to thank the reviewer for taking the time to elaborate on the CDD index study. Using the example of the Iberian Peninsula helped identify where our misunderstandings have arisen. We agree to eliminate any reference to 'frequency' as it has obviously lead to this misunderstanding. Both the reviewer and authors agree that the longest dry period of the three regions in Fig.6 is longer under 2.0°C than current conditions. The authors use of the term frequency was in regard to the changes in CDD distribution, which the reviewer has articulated ought to be written in terms of probability or likelihood, and as such the text has been changed (see L251, L254).

It should be noted that the CDD is calculated after a spatial mean of the Prudence region is taken and thus representative of the whole domain, not calculated per grid-box as the reviewer has written. This may have also lead to some of the misunderstanding. We have elaborated on the methodology as a result.

L253: I think correct English is "indistinguishable from" not "indistinguishable compared to"

A: Yes, changed as suggested.

L253-254: This is a ambiguous sentence. It's true that you can deduce that region 2, will suffer from more frequent and longer drought periods than experienced before. But compared to regions 6 and 8? Even if the increase in 2 is larger than in 6 and 8, the dry periods in 6 and 8 could still be longer than in 2 (I know it isn't, but still). I guess what you're after is that the longest dry period changes more in some regions (e.g region 2) and less in others (e.g. regions 6 and 8). L256-258: With the same kind of argument you could say that the British Isles would benefit from an temperature increase of 2 instead of 1.5. From Fig 6 its obvious that several regions would benefit from a lower temperature increase. Wouldn't you agree that the Iberian Peninsula would benefit more than the Mediterranean from a reduce warming? Although the change at 1.5 is already significant.

A: We have removed the term 'suffer from' and reformulated the sentence to state the facts.

L287: Please insert "RI50" somewhere here.

A: Added as suggested.

**Reviewer 2**

The current version clearly improves the paper. However, several parts are still not properly described/discussed, although several points were already raised during the previous review round. further revisions are necessary before your study can be considered for publication. Please refer to the main points and specific comments below.

A: We thank the reviewer for the comments, and we respond to the individual comments, below.

MAIN POINTS
* * *
1) HAPPI protocol: The description of the HAPPI approach is much improved. However, it still leaves out many aspects on the validity and limits of the approach, for instance with respect to the climate variability. This was already requested during the previous review process, but is not sufficiently addressed in the current version of the manuscript. For example, you could extend the Supplementary and include a comparison of the temporal variability (and not only the annual cycle) between the HAPPI simulations and observations.

A: We added a section in the discussion addressing this point, where we discuss the findings of Fischer et al. (2018) on AMIP vs. Coupled ensembles. This includes the aspects of atmospheric vs. oceanic variability and where we argue that our data should be mostly used for statistic based on daily or multi-daily data rather than seasonal or yearly.

2) Section 2.4:

2a) It would be worth a try to combine this section with the previous one (2.3). The reader would then have a direct overview of the individual indices (including calculation, changes and significance). Additionally, you could avoid the repetitive use of some phrases.

A: We agree with Reviewer#2 and merged the corresponding parts of section 2.4 with 2.3.

2b) Are the changes calculated from the ensemble mean or median?

A: This depends on the index. In case of ATG28 among others the 50$^{th}$ percentile is computed, which corresponds to the median. In case of RI50yr the relative difference is based on the mean. We added this information to the text.

2c) The calculation of significance is still unclear. There is no significance test for RI50yr, correct? The description for CDD is confusing: L177 states that differences are calculated similar to ATG28 with a Mann-Whitney U-Test, but you did not use this test for ATG28 but for RX5day... See also main comment 4h) of my first review.

A: For RI50yr no significance test has been done. We have changed this section in response to Reviewer#1 by deleting the confusing "similar to" statements. In addition, merging section 2.3 with 2.4 as suggested in comment 2a) should make clearer how each index and the corresponding change has been computed.

3) Discussion and conclusions: This part is still insufficient. In the previous round, the reviewers raised the important point on how your results relate/compare to other simulations (e.g. the MPI grand ensemble) and to other studies. Laura Suarez-Gutierrez even provided some examples. Nonetheless,

you do not consider any of these suggestions. There is not a single reference throughout the whole discussion and conclusion section. Instead, you argue that such a discussion is out of the scope of your paper. However, in order to provide convincing evidence to prove the benefits of your method this must not be neglected. It is even mandatory.

A: We agree that the discussion should include these examples. During the first round we misinterpreted Laura Suarez-Gutierrez comments and thought that we should include a full discussion on different GCM modelling approaches. We thank Reviewer#2 for this clarification and expanded the discussion and conclusion part accordingly.

4) Language: The language has improved significantly. Nevertheless, further revisions are needed. There are still some incomprehensible sentences, rather colloquial wording and superficial descriptions. Please also refer to the specific comments below.

SPECIFIC COMMENTS
* * *
1) References in the text should be sorted either chronologically or alphabetically (see e.g. L30f, L65).

A: We have double-checked the references. ESD also offers the option of sorting by relevance, which we followed in the specific case of L30. L65 has been split and in all other cases we followed chronological sorting (which often coincides with relevance anyway).

2) L11/12: "… for periods with levels of 1.5°C and 2.0°C global warming …" – Please reword.

A: Deleted "levels of".

3) L20/21: "… relevant to both the private and the public sector."

A: Changed "sectors" to "sector".

4) L32: "However, Mitchell et al. (2016) argue that …"

A: Changed accordingly.

5) L42-46: Please reword sentences and remove doubling of "bridge the gap between GCM model output and regional climate impact assessment".

A: We have rephrased this paragraph to:

"Regional climate impact assessments often require a much higher resolution than GCMs (e.g., Giorgi and Jones, 2009). To bridge this gap, dynamical downscaling with Regional Climate Models (RCMs) is one option, which provides physically consistent high-resolution climate information (Jacob et al., 2014; Giorgi and Gutowski, 2015; Gutowski et al., 2016). Here, the RCM REMO (Jacob et al., 2012) is used to dynamically downscale simulations from two GCMs of the HAPPI consortium. Two regional climate datasets of 25 and 100 members are developed, to create a large ensemble of RCM simulation, which are particularly suitable to study extremes."

6) L59: Could you add a reference for the IPCC Special Report?

A: Added as requested.

7) L59-61: Please reword sentence.

A: Section has been rephrased to:

"The HAPPI protocol by Mitchell et al. (2017) has been set up to inform the IPCC Special Report on 1.5°C Warming (IPCC, 2018). The idea is that large ensembles (>50 members) of GCM simulations will allow studying extreme events, even for the small differential warming between a current decade (2006-2015) and two future decades under 1.5°C and 2.0°C global warming."

8) L65/66: Reasoning unclear, please elaborate.

A: We rephrased the sentence to:

"All simulations were conducted in atmosphere-only mode in order to increase ensemble size, because of lower computational costs (Mitchell et al., 2017). In addition, He and Soden (2016) have shown that atmosphere-only mode simulations can provide more accurate regional projections, because they do not suffer from systematic biases such as SST drifts."

9) L71/72: "… temperature response for 2091-2100 compared to 1861-1880 is 1.55°C under RCP2.6."

A: Changed accordingly.

10) L75: "as" instead of "because"?

A: Changed accordingly.

11) L89-92: Could you include the difference between sponge zone and core domain in Figure 1?

A: Added to the plot.

12) L92-96: Please split sentence.

A: Changed accordingly

13) L128:/129: "… 1000 data points … 250 data points …" – Maybe include 100/25 members x 10 years?

A: Added information on members

14) Figure 1: Not every reader might be familiar with the PRUDENCE regions. Please explain abbreviations for individual regions in the figure caption. Some abbreviations are difficult read. Could you increase the readability?

A: We changed the caption of Figure 1 by adding the names of the PRUDENCE regions and improved readability inside the figure.

15) L172-174: Please split sentence.

A: Sentence has been rephrased in response to Referee #1

16) Caption Figure 2: "as" instead of "because"?

A: Changed accordingly.

17) L206: "are" instead of "is"

A: Changed accordingly.

18) Figure 4: The caption does not match the figure, you mixed up rows and columns.

A: We forgot to update the text after redoing the figure. The caption has been fixed.

19) Figure 5: The caption does not match the figure, you mixed up rows and columns. Additionally, there is a doubling of "in percent" and "in %".

A: We forgot to update the text after redoing the figure. The caption has been fixed.

20) L239-243: There is a doubling of the sentence "Where the p-value is greater or equal to …"

A: Sentence has been removed.

21) L239: "Where the resulting p-values are greater or equal to a significance level …" – The null hypothesis is that the distributions are the same, correct? Then it should be "Where the p-values are less or equal to a significance level…". Otherwise you marked the wrong numbers in Table 1.

A: Thank you for pointing this out. Indeed, it is "less than". This section has been rewritten on request of Reviwer#1.

22) Table 1: Why did you put the PRUDENCE regions in parentheses? "Hist" should be "current".

A: Changed as suggested.

**Reviewer 3**

The paper has already gone through one round of review and in particular the description of methodology is in much better shape now than the initial submission.

A: We thank the reviewer for the second review, and this positive comment, and we provide replies to the specific comments, below.

The duality of the paper trying to both presenting the data set and examples on how it can be used still detracts somewhat from both. Although I realize that this is likely beyond the scope of the paper I think the analysis would have been more interesting if the indices were also compared to observations and/or reanalysis.

A: As mentioned by the reviewer, a detailed performance measure of each individual index is beyond the scope of our paper as they should be seen as examples of what can be done with the data. In addition, we already added a supplement with some general performance measures to the supplement during the first round of revision.

The presentation of the data set itself can however be easily improved either through writing that you follow a certain standard, e.g. CORDEX and / or write a brief description in the supplement. What are the main variables and output frequency. Are they available for others? The availability notice says where they are stored, but not if they can be accessed.

A: Currently the data is available on request via the DKRZ cloud (swiftbrowser). We follow (as much as possible) the CORDEX and CMOR data conventions. We added a table with all available variables and frequencies to the supplement.

Since the resolution of the RCM is still quite coarse and not that different from the GCM (0.4 vs 1 degrees) I also think it would be useful if the authors discuss if they expect the conclusions to stay the same for higher RCM resolution. Although the pattern will likely be more noisy would e,g, 25 simulations with 0.22 degree resolution look like the 25 member subset? In particular many of the important meso-scale precipitation systems in the Mediterranean region are not resolved in a model with 0.4 degrees resolution.

A: For such large ensembles, there is always a trade-off between resolution, model complexity and ensemble size. In addition, many modelling groups in HAPPI did not have the resources to save the data that is necessary to perform dynamical downscaling. Given that ECHAM6 in the HAPPI experiments is relatively course (T63 or 1.875°), a direct downscaling to much higher resolution than 0.44° should we avoided, because the spatial spin-up of small-scale features inside the domain would become too large (see e.g., Matte, 2017). We agree that some of the important meso-scale systems are missed with 0.44°. Also the prescribed SST may not be appropriate to correctly simulate coastal precipitation especially around the Mediterranean and we briefly discuss this already in our paper and expanded on it (see below). To our best knowledge we do not expect fundamental changes in a 25-member 0.22° ensemble compared to our 0.44° subset. Details might change, but the qualitative patterns should look the same (if we neglect spatial spin-up issues discussed above).

We will add the following lines to the discussion part:

 "The relatively course resolution of 0.44° for a dynamical downscaling study over Europe is a compromise between model complexity, resolution and ensemble size. In addition, the course driving data particularly of the ECHAM6 model with T63 or 1.875° horizontal resolution does not allow a much higher resolution for direct downscaling without having a too large spatial spin-up of

small-scale features inside the domain (see, e.g. Matte, 2017). Besides details in some regions such as coastal precipitation around the Mediterranean, we do not expect fundamental qualitative changes to our findings on a higher resolution such as 0.22°."

line 3. "Dataset" is used two times in the same line

A: We changed the sentence to:

"The dataset is a unique and physically consistent, as it is derived from a large ensemble of regional climate model simulations."

line 70. anomaly → increment?

A: Changed accordingly

Line 90-95: Add information on GCM resolution.

A: ECHAM6 was run in T63 (1.875°) horizontal resolution and NorESM in 1.25°x0.94°. We added the resolution information to the text.

Line 97. How are the land use described in REMO for future scenarios?'

A: In REMO we followed the CORDEX approach and kept land-use constant for all periods. We added the information to the text.

"In REMO the same greenhouse gas forcings as for the GCMs were used and no land-use changes were applied."

Line 114. "one initial soil temperature state for every ensemble member in one period." But moisture profile is can vary?

A: We used this procedure for all soil variables. We deleted the word "temperature" from the sentence.

Line 139 "The index for the annual maximum. → minor quibble. Do you call it an index or can you just write The annual maximum …

A: Changed accordingly.

Figure 1: Perhaps obvious but can not see if IP is a subsection of MD (I presume it is)

A: We followed the original definition of the PRUDENCE regions where IP is not a subsection of MD.

line 215 "because the GCMs usually do not resolve these small basins. In these locations the SST is interpolated from the nearest SST value of the GCM, which might not be adequate for the region"

The basin is missing entirely? A larger grid size should just reduce the ocean fraction of the grid?

A: This refers to small sub-basins such as the Adriatic Sea. They are hardly resolved by the CMIP5 models and the question is, if such an SST is matching the regional characteristics. When looking at the HAPPI SSTs there is no West-East gradient detectable for the Adriatic Sea, because of insufficient resolution. We changed the formulation of that sentence and included a reference pointing out the importance of SST for heavy precipitation at the Adriatic Sea.

Matte, D.; Laprise, R.; Thériault, J. M. & Lucas-Picher, P.: Spatial spin-up of fine scales in a regional climate model simulation driven by low-resolution boundary conditions, Climate Dynamics, 2017, 49, 563-574